# Tuneable reflexes control antennal positioning in flying hawkmoths

Dinesh Natesan [1,2,3], Nitesh Saxena[1], Örjan Ekeberg [2] & Sanjay P. Sane[1]*

Complex behaviours may be viewed as sequences of modular actions, each elicited by specific sensory cues in their characteristic timescales. From this perspective, we can construct models in which unitary behavioural modules are hierarchically placed in context of related actions. Here, we analyse antennal positioning reflex in hawkmoths as a tuneable behavioural unit. Mechanosensory feedback from two antennal structures, Böhm's bristles (BB) and Johnston's organs (JO), determines antennal position. At flight onset, antennae attain a specific position, which is maintained by feedback from BB. Simultaneously, JO senses deflections in flagellum-pedicel joint due to frontal airflow, to modulate its steady-state position. Restricting JO abolishes positional modulation but maintains stability against perturbations. Linear feedback models are sufficient to predict antennal dynamics at various set-points. We modelled antennal positioning as a hierarchical neural-circuit in which fast BB feedback maintains instantaneous set-point, but slow JO feedback modulates it, thereby elucidating mechanisms underlying its robustness and flexibility.

[1] National Centre for Biological Sciences, Tata Institute of Fundamental Research, GKVK campus, Bellary road, Bangalore 560065, India. [2] Division of Computational Science and Technology, School of Electrical Engineering and Computer Science, KTH Royal Institute of Technology, 10044 Stockholm, Sweden. [3] Manipal Academy of Higher Education, Manipal 576104, India. *email: sane@ncbs.res.in

Animal movements such as walking, flying and swimming are composed of diverse modular behaviours, which require the nervous system to acquire and reliably encode ambient sensory cues. Each cue is transduced over different timescales, and the combined information is integrated by the nervous system to generate proper responses. This task is particularly challenging during fast movements when encoding and integration must occur on shorter timescales. For instance, for stable flight, insects need to integrate rapid mechanosensory feedback from their antennae with slower visual feedback from their compound eyes[1–3]. They actively position their antennae and eyes via fine feedback-control of antennal muscles and head/eye movements, thereby tuning sensory acquisition for flight stabilization[4,5]. Activity of sensory neurons also depends on the animal's internal state; for instance, activity of visual interneurons is different in moving vs. quiescent insects[6,7]. Such state-dependent modulation may enhance the functionality of antennae and eyes, perhaps by tuning their dynamic range.

From a systems perspective, how state-modulated sensory feedback influences positioning of sensory organs for optimal acquisition of sensory information poses a fascinating question. The insect antenna provides an excellent study system to address this question for several reasons. First, antennae are multi-modal probes which sense diverse olfactory and mechanosensory cues[8]. Thus, their movements may strongly depend on which cues insects are trying to maximize. Second, antennal movements are guided by multiple sensory cues including mechanosensation, vision and olfaction. Antennal position thus provides a convenient read-out to examine how nervous systems integrate sensory feedback from diverse modalities[9,10]. Third, antennal movements are context-specific, depending on whether the insect is walking, flying, foraging, escaping, etc[4]. Thus, state-dependent neuromodulation plays a key role in their control.

Smooth mobility of antennae over longer timescales must be balanced against the need for maintaining them in stable, unambiguous positions over shorter timescales. For instance, drifts in antennal position can alter mechanosensory feedback from Johnston's Organs (JO), which is essential for stable flight[2]. Maintenance of antennal position therefore requires one behavioural module to ensure smooth, unrestricted antennal movement and another to restrict its mobility. To examine how these counteracting modules control antennal position, we investigated antennal positioning in flying hawkmoths. Many insects, including hawkmoths, use mechanosensory feedback from Böhm's bristles to maintain stable antennal position[11–14]. Fields of Böhm's bristles located at the antennal base encode instantaneous antennal position relative to the head[11–16]. Ablating these bristles renders insects incapable of moving their antennae[12–14], underscoring their importance in antennal positioning. In diverse insects, the axons of the Böhm's bristles project into the antennal mechanosensory and motor centre (AMMC), where their arbours spatially overlap with dendritic arbours of antennal motor neurons[14,15]. This neural circuit allows swift antennal responses to perturbations while maintaining stable position.

On the other hand, slower modulation of antennal position is guided by multiple sensory cues. For instance, flying insects respond to frontal airflow by bringing their antennae symmetrically forward[9,17,18], whereas front-to-back optic flow induces antennae to move backwards, in the same direction as visual motion[9,19,20]. In walking insects, antennae track moving visual objects, but this behaviour is not bilaterally symmetric[21–23]. Odour cues also modulate antennal position[10,24]. Thus, the antennal motor system integrates information from several sources with disparate latencies, ranging from fast mechanosensory feedback from Böhm's bristles (<10 ms) to slower visual input from eyes (~35–60 ms)[14,19]. Although these cues are known to influence antennal positioning, the neural basis of this behaviour is not well-understood.

How does the antennal motor system integrate fast proprioceptive feedback at stroke-to-stroke timescales with relatively slower inputs from other sensory organs? Here, we address this question by investigating the neural principles underlying airflow-dependent antennal positioning in the oleander hawkmoth, *Daphnis nerii*. Because both stable positioning of antennae and airflow-dependent changes occur while preparing for or during flight, our experiments were performed on tethered flying hawkmoths. By simultaneously manipulating both antennal position and airflow, we recorded their antennal positioning responses to different airflows. Using a control theoretic framework, we compared antennal stability against varying airflows. These data allowed us to identify a minimal neural circuit which best explains how antennomotor reflexes are modulated in a robust, yet flexible manner. Thus, the antennal positioning response illustrates the hierarchical structure of behavioural modules and demonstrates how their mutual interactions enable fine control of this behaviour.

## Results

**JO mediate airflow-dependent antennal positioning**. Tethered flying moths were placed in a wind tunnel and presented with airflow stimuli of different magnitudes. We used two high-speed cameras (Fig. 1a, see the "Methods" section) to record changes in the inter-antennal angles (IAA) for a range of airflows (Fig. 1b). Experiments were carried out initially in an untreated (Control) group, followed by moths in which antennal mechanosensory feedback was restricted.

Two sets of mechanosensory structures, Böhm's bristles (antennal hair plates) and JO, are located on antennal scape and pedicel segments (Fig. 1c), which can actively move. Movements in head capsule–scape and scape–pedicel joints are sensed by scapal and pedicellar Böhm's bristles, respectively[11,14]. JO is composed of ca. 140 scolopidial sensory neurons, and senses passive vibrations in pedicel–flagellum joint[15]. Previous studies have shown that JO sense flagellar vibrations ranging from low-frequency vibrations due to airflow and gravity[25–27] to high-frequency flight-related movements of the antennal base[2,27,28].

We hypothesized that airflow-dependent antennal positioning is modulated by mechanosensory inputs from JO. To test this, we restricted motion in the pedicel-flagellar joint and monitored its response to variable frontal airflow. This treatment attenuated the passive flagellar vibrations that activate JO (Fig. 1c, see the "Methods" section), thereby disrupting mechanosensory feedback. Moths were divided into three groups: moths in the Control group were left unmanipulated, the Sham-treated group had the third/fourth annulus of the flagellum glued but pedicel-flagellar joint left free (Fig. 1c, Supplementary Fig. 1D), and JO-restricted group had pedicel-flagellar joint glued to reduce/eliminate mechanosensory feedback from JO (Fig. 1c, Supplementary Fig. 1F).

During tethered flight in control moths, IAA decreased as frontal airflow increased (Fig. 1d, Supplementary Fig. 2A, Supplementary Movie 1). This behaviour is henceforth termed as "airflow-dependent antennal positioning". Although initial IAA was different for each individual, it decreased monotonically as airflow varied from 0 to 3 m s⁻¹, and saturated when it exceeded ~3 m s⁻¹ (Fig. 1d, Supplementary Fig. 2A). The sham-treated moths behaved similarly to control moths (Fig. 1e, Supplementary Fig. 2B, Supplementary Movie 2).

In moths with restricted pedicel-flagellar joints, airflow-dependent antennal positioning was abolished during tethered

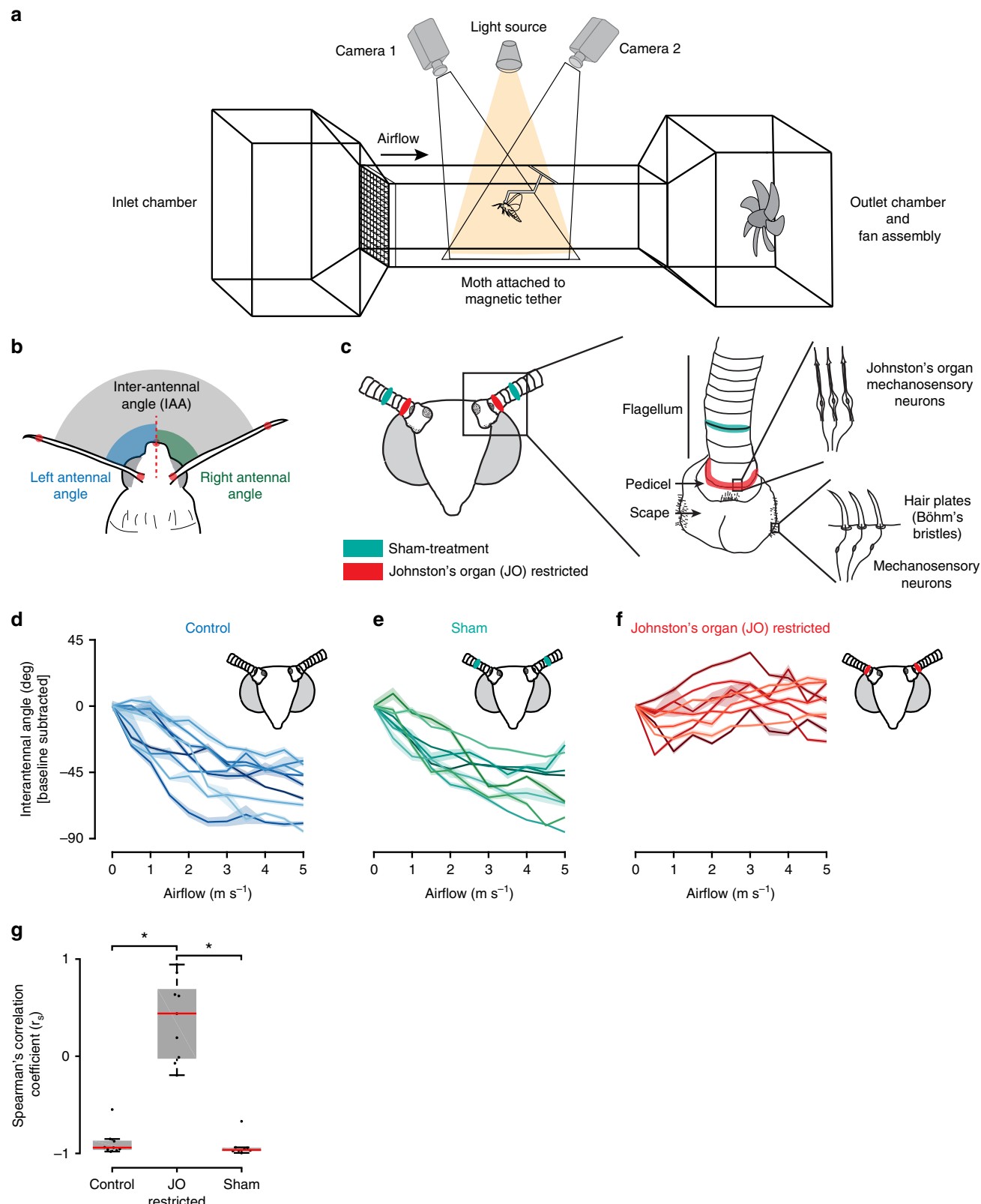

**a** Inlet chamber / Camera 1 / Light source / Camera 2 / Airflow / Outlet chamber and fan assembly / Moth attached to magnetic tether

**b** Inter-antennal angle (IAA) / Left antennal angle / Right antennal angle

**c** Sham-treatment / Johnston's organ (JO) restricted / Flagellum / Pedicel / Scape / Johnston's organ mechanosensory neurons / Hair plates (Böhm's bristles) / Mechanosensory neurons

**d** Control

**e** Sham

**f** Johnston's organ (JO) restricted

Interantennal angle (deg) [baseline subtracted] vs Airflow (m s⁻¹)

**g** Spearman's correlation coefficient ($r_s$) — Control, JO restricted, Sham

flight (Fig. 1f, Supplementary Fig. 2C–F, Supplementary Movie 3), although the antennae were still mobile (Supplementary Movie 5). Their IAA remained unchanged at low airflow values (Supplementary Fig. 2J, Kruskal–Wallis test, $p = 0.78$), and even increased slightly at higher airflow values, likely due to backward aerodynamic torques (Fig. 1f, Supplementary Fig. 2C).

The Spearman's rank correlation coefficient ($r_s$) between IAA and airflow (Fig. 1g, see the "Methods" section) was close to −1 for both control and sham-treated moths, implying that IAA decreased with increasing airflow. In contrast, $r_s$ ranged between 0 and 1 for JO-restricted moths, suggesting that antennal positions remained relatively unchanged, or increased with

**Fig. 1 Characterization of airflow-dependent antennal positioning. a** Experiment setup. Experiments were performed in a 1.2 m × 0.28 m × 0.28 m wind tunnel. The moth was tethered to the wind tunnel through a neodymium magnet glued to the thorax. Two high speed cameras filming at 100 fps captured the antennal responses during tethered flight. **b** Antennal angles. Inter-antennal angle (IAA) was computed as angle between antennae (grey). Head vector (dashed red line) was defined using head point and midpoint of antennal bases. Left and right antennal angle (blue, green) was computed as angle between respective antenna and head vector. **c** Treatment groups. JO-restricted: mechanosensory feedback from JO was restricted in a subset of moths by gluing the pedicel-flagellar joint (red). Sham-treated: To control for the effect of glue, annuli on the flagellum some distance from the pedicel-flagellar joint were restricted in another subset of moths (green). Note that active joints of the antenna were not glued, allowing full antennal mobility. **d–f** Antennal response to airflow during tethered flight. IAA of **d** Control (blue, $n = 9$), **e** Sham-treated moths (green, $n = 8$), and **f** JO-restricted moths (red, $n = 9$). The initial IAA was subtracted from the rest (baseline subtraction) to better visualize the response to airflow. Both Control and Sham-treated moths decreased their IAA in response to increasing frontal airflow, whereas JO-restricted moths maintained their antennae at constant angle. Different shades represent different individuals. Overlay around each line represents the standard error of the mean (s.e.m). **g** Statistical analysis of treatments. Box-and-Whisker plots of Spearman's correlation coefficient ($r_s$) of IAA response to airflow for all three treatments. The coefficient quantifies the monotonicity of IAA vs. airflow ($+1/-1$ perfect monotonic increase/decrease). Control and Sham-treated moths had negative coefficients suggesting a decrease in IAA with increase in airflow, whereas JO-restricted moths had $r_s$ ranging from 0 to 1, suggesting constant or slightly increasing angles (median – Control: $-0.94$, JO: 0.44, Sham: $-0.96$). The grey box represents central 50% data around the median (red line). The whiskers indicate data 1.5 times the interquartile range. Asterisks represent statistically different comparisons (Kruskal–Wallis, Nemenyi test, $p < 0.01$).

airflow (Fig. 1g, $p < 0.01$, Kruskal–Wallis test, Nemenyi test). Only restriction of JO input caused loss of airflow-dependent antennal positioning, implying that it requires feedback from JO.

**Conceptual model of airflow-dependent antennal positioning**. The results described above are also consistent with previous data in honeybees[9], indicating an evolutionarily conserved mechanism. Although mechanosensory inputs from JO control antennal responses to airflow, restricting these inputs by gluing pedicel-flagellar joint does not affect the ability of the animal to position antennae at flight-onset. Additionally, in honeybees, antennal response to other modalities, e.g. vision, is not affected upon JO-restriction[9]. Thus, initiation and maintenance of antennal position is independent of JO-mediated changes.

Along with sensory inputs from the JO, the antennal motor system receives proprioceptive inputs from Böhm's bristles[14,15]. Antennal movement stimulates neurons underlying these sensory hairs, in turn activating antennal motor neurons and associated muscles with latencies of ~10 ms[14]. Antennae are rendered immobile upon Böhm's bristles ablation, underscoring their importance in initiation and maintenance of antennal position[12–14]. Thus, mechanosensory inputs from Böhm's bristles are essential for maintaining stable antennal position. On the other hand, restricting inputs from JO by gluing the pedicel-flagellar joint does not affect the ability to position antennae, but disrupts airflow-dependent antennal movements (Fig. 1f, Supplementary Fig. 2C, Supplementary Movie 3).

These and previous results suggested a conceptual model of antennal positioning behaviour (Fig. 2a), where antennal position is encoded by Böhm's bristles which activate antennal muscles via a reflex arc[14]. This reflex operates as a negative feedback loop to ensure initiation and maintenance of antennal position during flight. JO-mediated airflow-dependent changes are achieved by modulation of the set-point (equilibrium position) of this negative feedback loop. Ablation of Böhm's bristles would break the feedback loop, disabling antennal positioning. On the other hand, restriction of JO inputs would cease airflow-dependent movements while still allowing initiation and maintenance of antennal position.

We propose that antennal-positioning behaviour comprises two hierarchically arranged sub-circuits (Fig. 2a); one maintains antennae at a preferred position or set-point using proprioceptive feedback from Böhm's bristles (antennal-positioning reflex) and the other modulates the set-point using sensory inputs from multiple modalities (set-point modulation circuit). Using an electromagnet set-up, we altered proprioceptive feedback from the Böhm's bristles by perturbing antennal position (Fig. 2b). We

could not ablate Böhm's bristles because removal of proprioceptive inputs renders the antennae immobile[14]. Inputs to the set-point modulation circuit were experimentally altered by changing airflow and restricting the JO (Fig. 1a, c). Therefore, using both experiments and computational simulations, we tested the hypothesis that airflow-dependent antennal positioning results from these two components working in concert, allowing its set-point to be tuneable.

**Airflow modulates set-point of antennal-positioning reflex**. Does airflow-dependent mechanosensory feedback from JO alter the antennal set-point? To test this, we perturbed the left antenna with an electromagnet at different values of frontal airflow while keeping the right antenna unperturbed as internal control (Fig. 2b, see the "Methods" section). Positions of both antennae were separately monitored, and each antennal angle was calculated with respect to the head vector to quantify the magnitude of electromagnetic perturbation (Fig. 1b, see the "Methods" section). Average position of right (unperturbed) antenna remained unaffected by magnetic perturbations delivered to the contralateral antenna (Fig. 2c, d, Supplementary Fig. 6). Thus, the reflex loop on each side of the antenna was local and independent of the contralateral antenna, consistent with previous findings that anatomical projections of Böhm's bristles do not cross the midline[14,15]. After removing the perturbation by switching off the electromagnet, the left (perturbed) antenna returned to the set-point corresponding to the current airflow value (Fig. 2e, f, Supplementary Fig. 6, Supplementary Movie 4). For greater values of frontal airflow, the antennae moved forward to smaller angles (Fig. 2g). Hence, antennal set-point is altered by frontal airflow and actively maintained during flight.

**Antennal-positioning reflex is unaffected by lack of JO inputs**. If mechanosensory feedback from JO modulates only set-point of antennal motor neurons, then restricting it should not affect antennal-positioning reflex. To test this, we restricted JO by gluing the pedicel-flagellar joint and magnetically perturbing antennal position. These moths did not show airflow-dependent antennal positioning (Fig. 3a, b, Supplementary Fig. 6), but nevertheless corrected for perturbations (Fig. 3c, d, Supplementary Fig. 6) and maintained antennal position at initial set-point (Fig. 3e, Supplementary Movie 5). Antennal set-points for JO-restricted moths remained constant for low airflow values and, in some cases, increased for higher airflow (4 m s$^{-1}$), likely due to aerodynamic drag. Spearman's rank correlation coefficients ($r_s$) between set-points and airflow was significantly

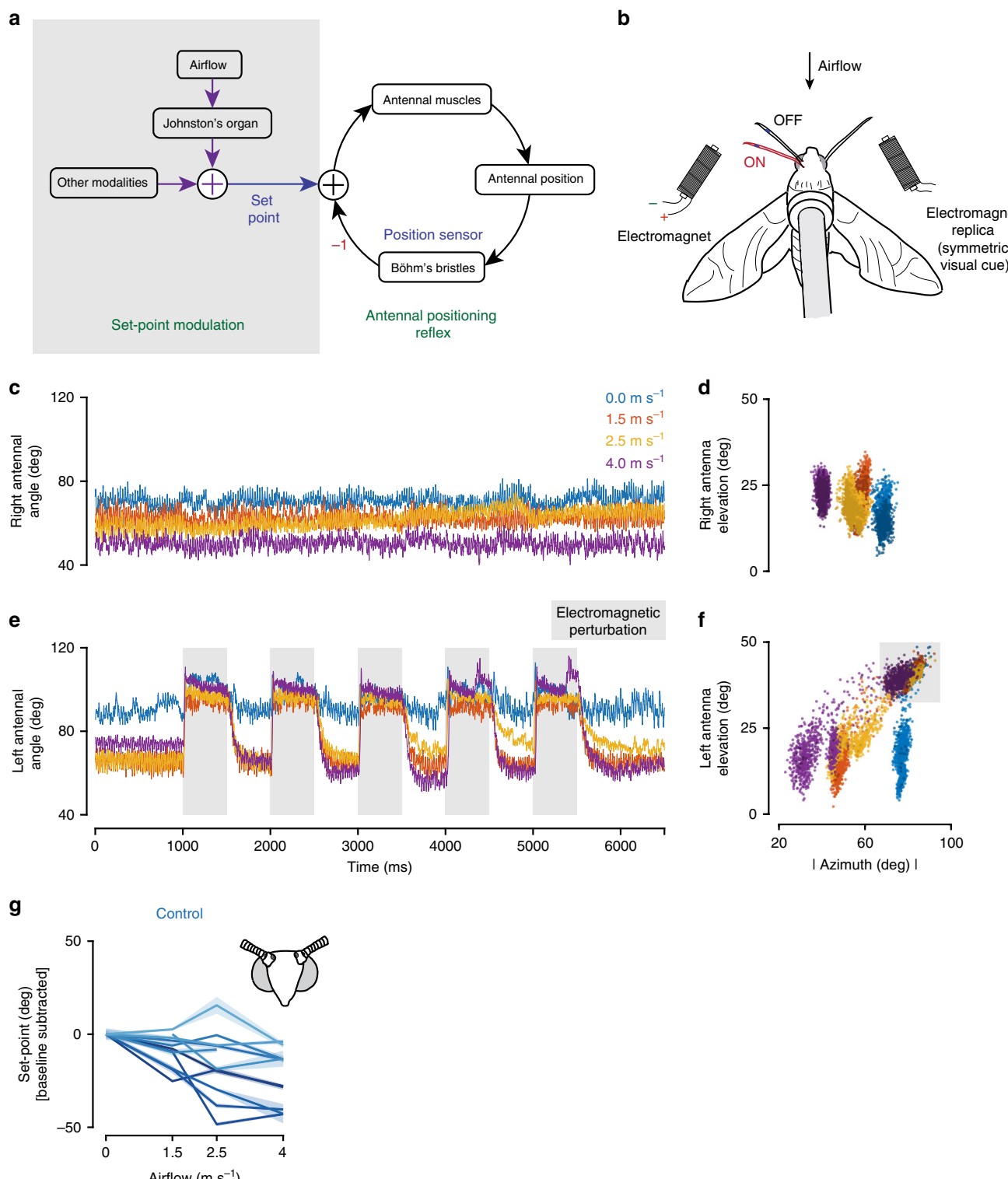

different for control and JO-restricted moths (Fig. 3f, $p < 0.01$, Wilcoxon rank sum test). These data show that JO-restriction does not alter the ability of moths to maintain antennal position, but disrupts their ability to modulate set-point based on frontal airflow.

**Simple linear models approximate antennal-positioning reflex.** We next developed a control theoretic framework to compare the dynamics of antennal-positioning reflex at different set-points. We expressed the conceptual framework (Fig. 2a) as a

linear control model, with the transfer function $L(s)$ representing error correction dynamics of the antennal-positioning reflex (Fig. 4a, upper panel). $L(s)$ was represented as a standard set of transfer functions including Proportional (P), Integral (I), Proportional-Integral (PI), Proportional-Differential (PD), Double-integral (II) and Proportional-Integral-Differential (PID) systems (Fig. 4a, lower panel, see the "Methods" section). These models were fit to the return trajectory of a perturbed antenna using the System Identification Toolbox in MATLAB.

**Fig. 2 Electromagnetic perturbation of antennal position in flying hawkmoths. a** Conceptual model for airflow-dependent antennal positioning. The model comprises two sub-circuits. "Antennal-positioning reflex" is a fast sub-circuit that maintains antennal position at set-point (preferred position) using proprioceptive feedback from Böhm's bristles. "Set-point modulation circuit" continually modulates the set-point based on airflow, via the JO. **b** Experimental setup for electromagnetic perturbations of antennal position. Electromagnets were used to perturb the antennae in order to quantify stability at different airflows. Iron filings were glued to the left antenna and perturbed during tethered flight using the left electromagnet (the right electromagnet was retained for visual symmetry but otherwise not used). The response was filmed at 1000 fps using the same cameras shown in Fig. 1a for four different airflows. **c–f** Response to perturbations in control moths. Representative raw data plots of antennal response to perturbations. **c** The right antenna (internal control) was unaffected by the perturbations to the left antenna, and its position depended only on the frontal airflows. **d** Azimuth-elevation plots show the clustering of right antennal position based on airflow. **e** When the electromagnet was on (grey), the left antenna was perturbed to a different angle, which was actively corrected on electromagnet release. The corrected angle depended on the frontal airflow. Sometimes the moths varied their corrected position during trials. An example of this is the response of the representative control moth at $1.5\,\mathrm{m\,s^{-1}}$. Such changes may arise due to modulations in set-point owing to other modalities (Fig. 2a). **f** Five distinct antennal position clusters were observed, of which four corresponded to the subjected airflows, and the fifth to the perturbed location. **g** Antennal set-points of control moths for different airflows. Set-points (corrected positions) of control moths decreased with increasing airflow. Different shades indicate different trials ($n = 11$ trials from 9 moths). The overlay indicates the standard error of the mean (s.e.m.).

Two measures were used to quantify the goodness-of-fit of transfer functions: adjusted $R^2$ and normalized Akaike information criterion (nAIC). Coefficient of determination ($R^2$) estimates the percentage of variation in the raw data explained by a model. nAIC quantifies the information lost when a model is used instead of raw data. Both measures penalize models with high number of parameters (i.e. complex models). The model with the best fit would, therefore, have a high adjusted $R^2$ and low nAIC. These measures, when combined, provided a reliable estimate for goodness-of-fit of transfer functions.

Transfer functions I, PI, PD and PID best fit the error correction dynamics underlying antennal return trajectories based on both measures (control: Fig. 4b–e, JO-restricted: Supplementary Fig. 3A–D; parameters given in Table 1). Goodness-of-fit of these models were statistically different from other models (P, II), but not from one another (control: Fig. 4d, e, JO-restricted: Supplementary Fig. 3C, D). Simple linear models, such as Integral system (Fig. 4c, Supplementary Fig. 3B), can hence closely the fit error correction dynamics of antennal-positioning reflex.

**Error correction dynamics are independent of set-points.** Antennal positioning reflex is operational even as antennal set-point varies with different frontal airflows, and it is also active when feedback from JO is severely reduced. This suggests that error correction dynamics of the antennal-positioning reflex are independent of the actual set-point. To test this, we quantified how accurately the models predicted the return trajectory at different airflows based on dynamics from one airflow.

Transfer functions I, PI, PD and PID with constants based on single airflow values reliably predicted the dynamics of all airflow values (control: Fig. 4f, Supplementary Fig. 3E, F; JO-restricted: Supplementary Fig. 3G–I). The goodness-of-fits of these predictions were equivalent to fitting the transfer function on the experimental data (compare Fig. 4d, e, Supplementary Fig. 3E, F; Supplementary Fig. 3C, D, H, I). Simple linear models can thus closely predict the error correction dynamics of the antennal-positioning reflex for all antennal set-points. Additionally, the consistent dynamics in error correction, irrespective of set-point, suggest an active rather than passive mechanism. These results support a simple linear feedback loop with an adjustable set-point as the underlying neural circuit for antennal positioning.

**Neural circuit for airflow-dependent antennal positioning.** The above experiments and control theoretic analyses showed that airflow-dependent antennal positioning arises from an interplay between the antennal-positioning reflex and a circuit that modulates the set-point. Additionally, the antennal-positioning reflex

can be modelled as simple linear models which both fit and predict their return dynamics irrespective of set-point. These models formally describe and characterize the computations underlying airflow-dependent antennal positioning, but do not provide a mechanistic explanation of how neural circuits perform these computations. We therefore proposed a minimal neural circuit that incorporates the simple linear models described above and simulated it as a feasibility test. Because a group of linear models (I, PI, PD, PID) fit and predict the antennal return dynamics equally well, we modelled the minimal neural circuit as an integral model on the basis of parsimony (Fig. 4a).

The minimal neural circuit is based on electrophysiological and neuroanatomical data from previous studies, which showed that mechanosensory neurons underlying Böhm's bristles activate antennal motor neurons, likely via direct connections[14,15]. Therefore, in the minimal circuit, mechanosensory neurons were treated as simple on–off neurons that monosynaptically activate antennal motor neurons (Fig. 5a, see the "Methods" section). On the other hand, mechanosensory inputs from JO do not appear to form synapses with motor neurons[15]. Hence, in our minimal circuit model, we assumed that they connected via interneurons (Fig. 5a, see the "Methods" section). Finally, we modelled motor neurons as simple integrate-and-fire neurons that pool incoming activity and control antennal muscles, and thus also the antennal position (Fig. 5a, see the "Methods" section).

As a template for the above neural circuit, we used the scape–pedicel joint with only one degree of freedom to simplify the model mechanics. As in the actual case, the model pedicel has two proprioceptive Böhm's bristles fields that sense its movement relative to the scape, and two muscles that control this motion (Figs. 1c and 5a)[29]. This minimal circuit can correct position and maintain it in response to simulated antennal perturbation (Fig. 5b, Supplementary Fig. 4A, B). The set-point of this circuit could also be modulated by changing firing rates of the interneuron carrying information from JO (Fig. 5c, Supplementary Fig. 4A, B). It maintained position at the modulated set-point despite external perturbations (Fig. 5c, qualitatively similar to Fig. 2E).

We further analysed the neural circuit using an identical control framework as in the behavioural data, and found that its dynamics were recapitulated by I, PI, PD and PID systems (Fig. 5d–f, Supplementary Fig. 4C; parameters given in Table 1). PID fit the data better than I (median: 1.00, 0.97, respectively), possibly because it also fit the noise due to the Poissonian nature of model neurons (Fig. 5f, see the "Methods" section). Both models captured the return dynamics for all simulated set-points (Fig. 5f, Supplementary Fig. 4C). Moreover, they predicted the return dynamics at all set-points based on the dynamics of any one (Supplementary Fig. 4D–F). Thus, the realized neural circuit

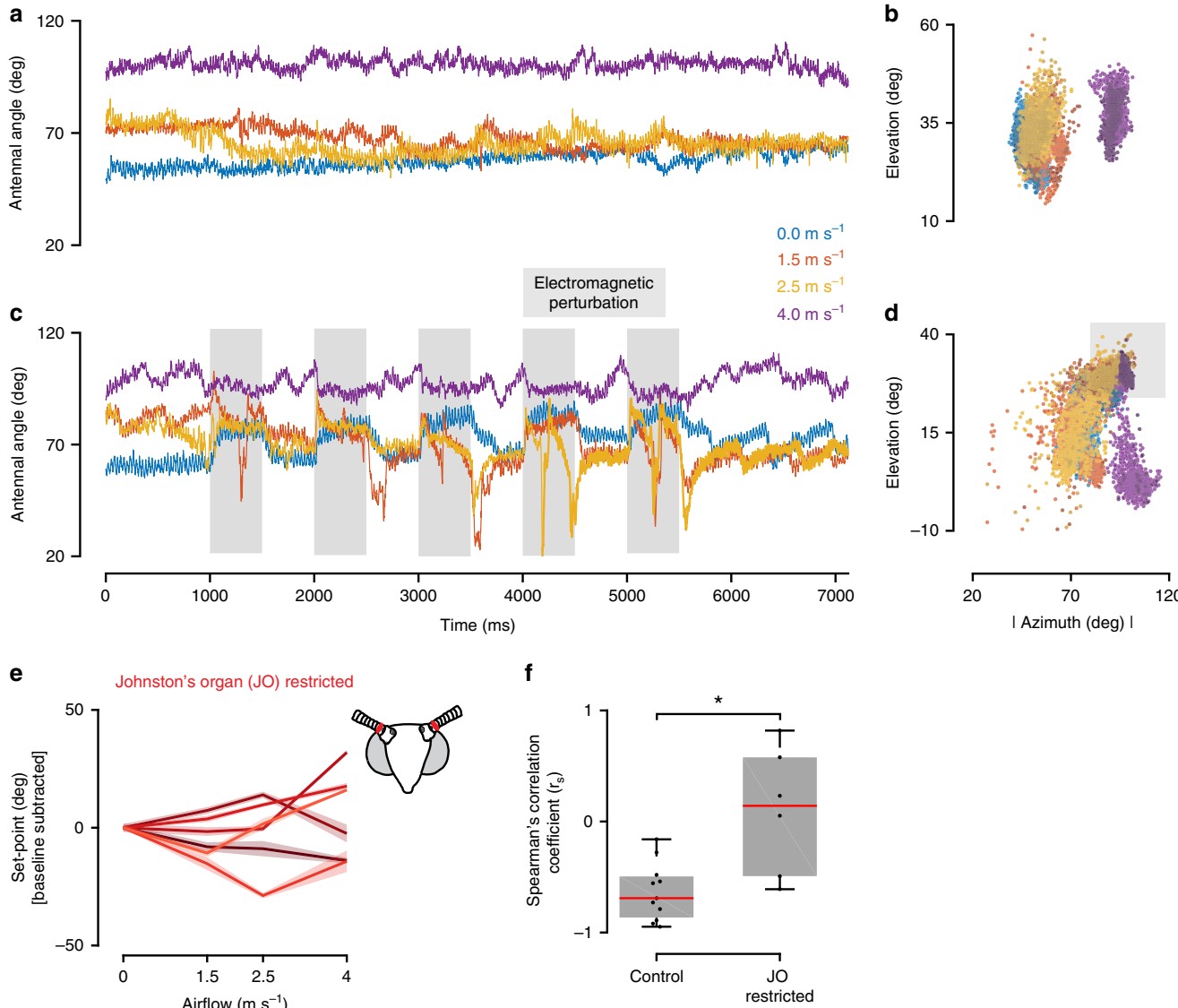

**Fig. 3 Electromagnetic perturbation of antennal position in JO-restricted moths.** **a–d** Representative raw data plots of antennal response to perturbations in JO-restricted moths. **a** The right antenna (internal control) was unaffected by the perturbations of the left antenna, same as in control. **b** Azimuth-elevation plots show the clustering of right antennal position based on airflow. The clusters for 0, 1.5 and 2.5 m s$^{-1}$ were identical. At higher airflows (4 m s$^{-1}$, purple trace) the antenna was unable to maintain the same position and drifted slightly backward. **c** The perturbed antenna was actively corrected in JO-restricted moths. They, however, were corrected to the same position regardless of airflow (except for 4 m s$^{-1}$). The sudden spikes in between were caused by the movement of the antenna by the ipsilateral front leg. **d** Three distinct antennal position clusters were observed—two based on airflow (0, 1.5 and 2.5 m s$^{-1}$ form one cluster and 4 m s$^{-1}$ forms the other one); the third is the perturbed location. **e** Antennal set-points of JO-restricted moths. JO-restricted moths also correct their antennae, but the corrected position (set-point) remains constant regardless of the frontal airflow. Different shades indicate different individuals ($n = 6$ individuals). The overlay around each line indicates the standard error of the mean (s.e.m.). **f** Statistical analysis. Box-and-Whisker plots of Spearman's correlation coefficient ($r_s$) of set-points vs. airflow for two treatments (control: $n = 11$ trials, 9 moths, JO: $n = 6$ trials, 6 moths). Change in set-point in response to airflow was significantly different between control and JO-restricted moths (asterisks represent statistical difference, Wilcoxon rank sum test, $p < 0.01$; median values—Control: −0.69, JO: 0.14).

maps onto the control theoretic model, providing a mechanistic basis for airflow-dependent antennal positioning.

## Discussion

Antennal positioning in moths consists of two behaviours operating at different timescales. The first is antennal positioning at flight onset and its maintenance during flight, which requires error corrections at stroke-to-stroke timescales (antennal-positioning reflex). The second (stimulus-dependent antennal positioning) involves slower positional modulation based on multisensory

inputs like optic flow, airflow and odour. Whereas latency of the antennal positioning reflex is typically < 10 ms[14], latencies of modulatory inputs are longer (e.g. optic flow: 35–60 ms[19]). Because antennae are crucial in sensing olfactory, mechanosensory, hygrosensory and thermosensory stimuli, control and maintenance of their position is critical.

Antennal positioning at flight onset and its airflow-dependent modulation have been observed in diverse insects (honeybees, locusts, flies[17,18,20]). Airflow-dependent modulation is mediated by the JO, which likely senses antennal deflections due to aerodynamic torques[4,30,31]. How this behaviour aids in sensory

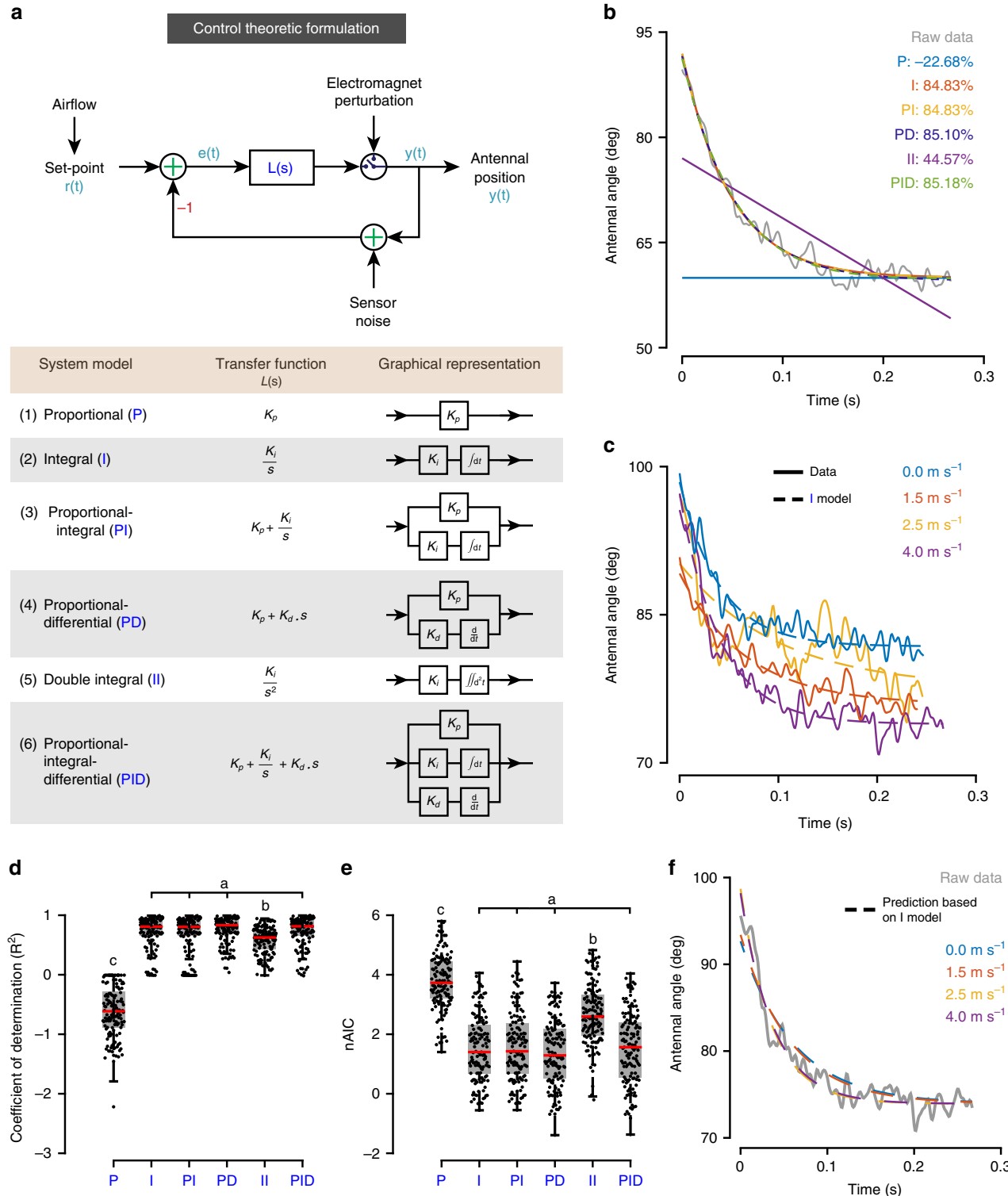

acquisition is as yet unclear; airflow-dependent modulation does not seem to maintain aerodynamic torques (Supplementary Fig. 2G–I, also refs. [30],[31]), and depends on multiple sensory inputs including optic flow[9].

We investigated the mechanisms underlying control and maintenance of antennal position and developed a control theoretic and neural circuit model to account for the observed behaviour. At the core of these models is a fast, linear, negative feedback loop that reflexively maintains antennae at a fixed set-point on stroke-to-stroke timescales. Overlying this feedback loop

are slower, modulatory influences due to frontal airflow, and presumably other modalities that alter this set-point in proportion to the appropriate sensory input. When these modulatory inputs are reduced (e.g. by restricting JO), moths retain their ability to maintain antennal position at an arbitrary set-point, but do not alter the set-point at which the antennae are held (Fig. 3). For instance, JO-restricted honeybees retain the ability to modulate antennal position based on optic flow[9]. This suggests the existence of an intrinsic set-point for antennal position in the absence of other sensory cues. In the presence of other sensory

**Fig. 4 Control theory models of airflow-dependent antennal positioning. a** Control theoretic formulation. To analyse the return characteristics of the antenna, we reformulated Fig. 2a using a control theoretic framework. The model takes in set-point as input, which is modulated by airflow via JO. Output is antennal position, which is sensed and fed back by Böhm's bristles. The error in position is convolved with a transfer function to obtain the output. We used six transfer functions—Proportional (P), Integral (I), Proportional-Integral (PI), Proportional-Differential (PD), Double-Integral (II) and Proportional-Integral-Differential (PID). **b** Model fits on antennal return trajectory. A representative return trajectory of the antenna is plotted in grey, and model fits are plotted in colour. Out of the six models, I, PI, PD and PID fit the representative antennal return trajectory well (>80% fit). The integral model (I) was the most parsimonious based on system components. **c** Integral model fits for a representative control dataset. The integral model fits the return trajectories of all airflows (shown for one representative dataset). The solid line represents raw data, with its colour representing the airflow for the return trajectory. The dashed lines represent the integral model estimations. **d**, **e** Goodness-of-fit of models. Box-and-whisker plots of **d** coefficient of determination ($R^2$) and **e** normalized Akaike information criterion (nAIC) for all system models. Fits based on I, PI, PD, PID models were significantly different from the rest (a, b, c represent statistically different groups, Kruskal Wallis, Nemenyi test, $p < 0.01$; $n = 133$ trajectories; median values of $R^2$—P: −0.61, I: 0.81, PI: 0.80, PD: 0.83, II: 0.63, PID: 0.81; median values of nAIC—P: 3.73, I: 1.40, PI: 1.43, PD: 1.29, II: 2.59, PID: 1.56). **f** Predictive capabilities of the integral model. A representative return trajectory of the antenna at an airflow of $4\,m\,s^{-1}$ is plotted in grey. The dashed lines represent predictions of integral models generated based on trajectories from only one airflow (indicated by their colours). All such integral models predict the return trajectory of the antenna, thereby suggesting that the dynamics of error correction are independent of set-point.

### Table 1 Fitted parameters of control theoretic models.

| | P—$K_p$ | I—$K_i$ | PI—$K_p$ | PI—$K_i$ | PD—$K_p$ | PD—$K_d$ | II—$K_i$ | PID—$K_p$ | PID—$K_i$ | PID—$K_d$ |
|---|---|---|---|---|---|---|---|---|---|---|
| Control | 277,183.36 ± 12,165.88 | 22.50 ± 1.71 | 40.60 ± 5.45 | 543.07 ± 27.75 | 81,856.67 ± 15,559.45 | 3363.48 ± 510.77 | 31.65 ± 7.62 | 12.32 ± 1.22 | 50.15 ± 15.91 | 1.84 ± 0.39 |
| JO-restricted | 183,833.13 ± 17,525.61 | 21.80 ± 2.70 | 33.16 ± 3.43 | 485.98 ± 48.47 | 67,682.04 ± 20,216.36 | 3175.06 ± 733.58 | 97.59 ± 28.30 | 10.27 ± 1.42 | 89.32 ± 45.05 | 3.16 ± 1.61 |
| Neural circuit with intrinsic setpoint | 184,733.75 ± 19,417.91 | 13.88 ± 0.27 | 15.47 ± 1.44 | 218.90 ± 19.20 | 66,715.73 ± 7703.52 | 4677.09 ± 503.23 | 0.04 ± 0.00 | 4.05 ± 0.55 | 55.32 ± 9.30 | 0.22 ± 0.02 |
| Neural circuit with modulated setpoint | 458,754.41 ± 5584.12 | 11.91 ± 0.11 | 36.24 ± 1.34 | 440.92 ± 15.44 | 7.97 ± 1.06 | 1.16 ± 0.09 | 0.09 ± 0.01 | 3.39 ± 0.41 | 22.45 ± 3.47 | 0.40 ± 0.04 |

Mean fitted parameter values for each of the control theoretic models along with the standard error of the mean (s.e.m.)

cues however, the set-point is modulated to a new value (Fig. 2c–g, also ref. [9]). Although airflow-dependent changes appear to symmetrically modulate set-points of both antennae, this may not be true for other modalities. Indeed, asymmetric responses of antennae have been observed in previous studies on optic flow and odour[10,19–24], perhaps due to unequal modulation of set-points of local reflex loops.

To further illustrate the effect of set-point modulation on the antennal-positioning reflex, we modelled error correction dynamics of the perturbed antenna using control theoretic methods. Because the precise details of the muscle and bio-mechanical properties of the antennal motor system were not known to us, we modelled the entire antennal circuit as a single system, which includes both the neural controller and bio-mechanical plant. Although our model nomenclature resembles neural controller models in literature[32–34], these are in fact systems-level transfer functions.

We used standard linear models to fit error correction dynamics and found that they depended only on the error between current position and set-point. Dependence on error in current position instead of absolute position indicates an underlying mechanism of active error correction rather than passive mechanical rebound. The linear control theoretic model $L(s)$ captures dependence of output position with respect to this error, with the integral model being mathematically equivalent to a decaying error exponential. Other higher-order models (PI, PD, PID) can capture more complex dependencies between output and error. We provided different amplitudes of step perturbations, but this disturbance was insufficient to distinguish differences in performances of higher-order models. Other stimuli (e.g. sum of sines, white noise or chirps) may provide better resolution on which of the four models (I, PI, PD, PID) best approximates error correction dynamics of the antennae for a variety of disturbances[35–37].

Our alternate hypothesis was a non-linear model in which error correction dynamics depend on antennal set-point. Such dynamics could occur if modulatory inputs like optic flow or airflow alter not just the set-point, but also the time constants of control models. Such a system can maintain the antenna at set-point, but the error correction dynamics would change as the set-point changes. In such scenarios, predicting antennal return dynamics at different set-points would not be possible based on just one set-point. To differentiate the above scenario from the linear model (Figs. 2a, 4a), we quantified how well these models could predict antennal dynamics in other airflows based on dynamics of just one case. Such predictions were only possible in the linear case, for which the underlying dynamics did not alter based on set-point. The predictions explained a large range of return trajectories in all airflows (median of 0.76 for I model, Supplementary Fig. 3), suggesting that the linear model was sufficient.

In our control model, set-point was assumed to be fixed for each value of airflow. This allowed us to isolate and characterize the inner loop dynamics, thereby identifying its stereotypic error correction (Fig. 4a). Because goodness-of-fits for these models were high (Fig. 4d, e), we inferred that inner loop timescales were faster than those of the set-point modulation circuit. We concluded that the Böhm's bristle-mediated reflex loop rapidly maintains antennal position, whereas feedback from JO slowly modulates set-point based on airflow. However, airflow sensing by JO and resulting set-point modulation have their own temporal dynamics, which may interact with those of the inner feedback loop during flight. This may be especially true in variable airspeed conditions, e.g. rapid flight manoeuvres or a sudden wind gust. This motivates the need for experiments and modelling specifically targeted towards understanding how insect nervous systems disambiguate the interacting timescales of these two circuits.

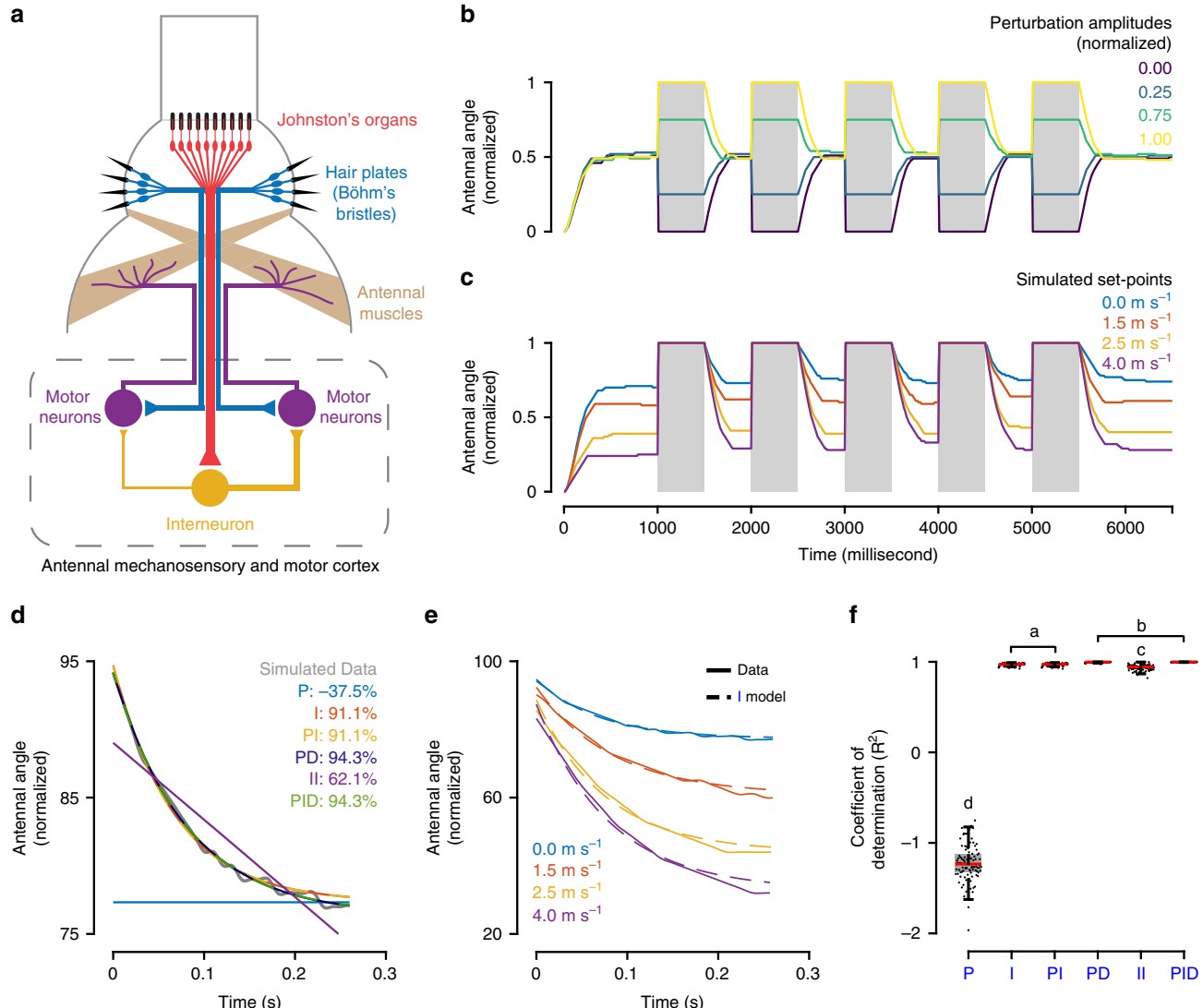

**Fig. 5 A model neural circuit for airflow-dependent antennal positioning. a** Model neural circuit. Motor neurons summate activity from mechanosensory neurons from Böhm's bristles and interneurons transmitting sensory inputs from the JO. The connectivity between the motor neuron and the muscles give rise to the negative feedback in Fig. 2a; motor neurons activate muscles which, upon contraction, reduce feedback from the hair plates, thereby decreasing their own activity. Muscles, due to the slow calcium integration times, integrate error in position. Sensory inputs from JO asymmetrically activate the motor neurons, thereby modulating antennal set-point. **b** Simulated antennal positioning reflex. Simulation of the model in Fig. 4a using NEST simulator without any set-point modulation. The simulation protocol was the same as in experiments. Simulated antennae corrected their positions to the intrinsic set-point of the neural circuit. Different colours represent different levels of perturbation. **c** Set-point modulation of the antennal positioning reflex. Set-point was modulated by asymmetric excitation of the motor neurons. The simulated antennae were corrected and maintained at different positions based on excitation by the interneuron. Different colours represent different modulations of the intrinsic set-point. **d–f** Control theoretic analysis of the simulated antenna. **d** Fits of all six models on the return trajectory of the simulated antenna. Four models—I, PI, PD, PID—fit the representative simulated antennal trajectory well (>80% fit). The integral model (I) was the most parsimonious of them. **e** Fits of the integral model on return trajectories for different set-points. **f** Box-and-Whisker plots of coefficient of determination ($R^2$) for all system models (a-d represent statistically different groups, Kruskal–Wallis, Nemenyi test, $p < 0.01$; $n = 100$ trajectories; median values of $R^2$—P: −1.23, I: 0.97, PI: 0.97, PD: 1.0, II: 0. 94, PID: 1.0).

As a first step towards addressing these requirements, we proposed a minimal neural circuit model that could maintain and modulate position based on sensory stimuli. The neural circuit model was built on the linear models and existing anatomical and physiological data[14,15]. Because a whole group of models were able to fit and predict the error correction dynamics of the antennal-positioning reflex, several neural circuits are possible. The integral model was used based on parsimony of components and formed the basic framework for higher order models. We used this control theoretic approach to generate mechanistic hypotheses of underlying neural circuits that perform this computation.

We decoupled contributions of the neural circuit (controller) from the biomechanical system (plant), based on the time constant of the integral system. The integration constants of the I model were in the same range as those of muscle calcium dynamics[4,38] (Table 1). We therefore assumed that muscles perform this integration in the neural circuit model, making them the biomechanical plant. Estimating the error between set-point and current position was the only remaining computation, which could be performed by the neural controller. Computing the error requires a difference operation, either due to inhibition or antagonistic excitation. We chose antagonistic excitation over inhibition due to the prevalence of excitatory neurons in the

antennal motor system[4]. Output of a neural circuit with the above architecture matched observed behaviours (compare Fig. 5c with Fig. 2e).

The model neural circuit captures the information flow from Böhm's bristles and JO to the antennal muscles. Böhm's bristles control motor neuronal activity via negative feedback (inhibition/antagonistic excitation), whereas JO modulates its set-point. The specific set-point modulation can happen in multiple ways, including activation of antennal motor neurons by JO either via interneurons or by direct synapses onto them. However, the latter possibility is not supported by neuroanatomical data, which show very little co-localization between axonal arbors of JO with dendritic fields of Böhm's bristles[15]. This modulation translates to asymmetric activation of antennal motor neurons by JO, which is a key feature of information flow from JO in the model neural circuit (Fig. 5a).

Here, the only major computation performed by the neural controller is summation. The motor neuron was modelled as an integrate-and-fire neuron that integrates synaptic inputs (EPSPs) to fire spikes. Because of this membrane voltage integration at fast timescales, the output firing rate is proportional to input firing rate. Mathematically, the computation performed by the motor neuron is equivalent to summation of all its inputs and multiplication by a gain. This linear computation makes the neural controller extremely fast. Such linear response characteristics are a feature of higher order fast behaviours such as flower tracking in moths[36,39] and wall-following in cockroaches[34], suggesting that this may be a general feature for fast behaviours.

The intrinsic set-point of our model neural circuit (i.e. position in the absence of any modulatory sensory inputs) results from both the mechanical placement of bristles and muscles, and the underlying sensory to motor neuron connectivity. In absence of modulation, equal activation of both hair plates results in simultaneous muscle activation on both sides (Fig. 5a, b). JO feedback can modulate this set-point by asymmetrically activating the motor neurons without changing error correction dynamics (Fig. 5c). Altering error correction dynamics requires changing the overall excitability of both muscles, which in turn requires symmetric excitation/inhibition of all motor neurons/muscles. For example, in crickets, inhibitory or excitatory dorsal unpaired median (DUM) neurons symmetrically innervate antennal muscles on both sides[4,40–42]. Activity in these neurons may change dynamics without altering set-point. Such changes might explain the slow return to baseline seen in a few trials (Fig. 2e, Supplementary Fig. 6).

Can this model be extended to other systems? The reflexive activation of motor neurons due to mechanosensory feedback, the basis of the model neural circuit for the antennal motor system, is also observed in other systems. For example, wing stretch receptors in locusts monosynaptically activate wing depressor motor neurons while simultaneously inhibiting wing elevator motor neurons[43]. Stretching the wing during an upstroke activates muscles that initiate a downstroke, thereby indirectly reducing wing stretch. Similarly, hair plates in certain segments of insect legs, which are likely precursors of the Böhm's bristles[44], sense changes in joint angles and directly activate muscles that reduce these angles[45–48]. These examples are analogous to activity of Böhm's bristles in the model proposed here (Fig. 5a), in which antennal position is stabilized via antennal muscles, but with the additional attribute that it is tuneable. In such feedback loops, the intrinsic set-points depend on mechanosensory feedback and its influence on the associated motor neuron, whereas modulation of the set-point may be controlled by interneurons synapsing onto this motor neuron (Fig. 5a). The model neural circuit, therefore, provides a functional hypothesis for modulation of rapid reflexes with similar connectivity.

Antennal positioning, as examined in this study, is an example of state-dependent behaviour in flying insects. Antennal position impacts acquisition of both mechanosensory and olfactory cues. For instance, active forward movement of antennae with increasing airflow may help restrict the flagellum to operate in the linear range of the pedicel–flagellar joint, thus enabling reliable acquisition of airflow-related or other flagellar vibrations[30,31]. However, this may diminish the ability to sample the odour space around the insect as it decreases spatial sampling[49,50]. On the other hand, if antennae are held at large angles to increase odour capture, aerodynamic drag increases, affecting its mechanosensory function[30,31]. This sets up a potential trade-off in which increasing sensitivity of one (olfactory) cue compromises sensitivity of the other (mechanosensory). In addition to mechanosensory and olfactory cues, antennal position is also influenced by visual feedback[9,19,20], which may impact both olfactory and mechanosensory feedback. Such trade-offs for acquisition of sensory stimuli have received very little attention.

The linear integration model proposed here suggests a specific mechanism by which multiple cues may be integrated by antennal motor neurons. A tuneable reflex (Fig. 5a) ensures that antennae have the necessary flexibility to integrate information from diverse cues, while ensuring that its position is sufficiently stable to reliably acquire information. The intensity or requirement of each cue would then determine new set-points for the antennae, while the Böhm's bristles mediated antennal-positioning reflex would ensure set-point maintenance despite external perturbations. Because the antennal-positioning system and associated neural circuits are fairly well-conserved[51], the tuneable reflex proposed here may be used to understand antennal movements in diverse insects.

In summary, fast behaviours in insects are often considered reflexive, and in neural terms sparsely connected or hard-wired. From a controls standpoint, such reflexes may be formally evaluated as negative-feedback loops. Whereas a reflex circuit ensures rapidity of responses, it precludes the variability often required to tune the response in a context-dependent manner. Such fine-tuning in response to multiple inputs, such as olfaction and vision can be slower and controlled by accessory feedback loops. In this paper, we show that a seemingly simple reflexive response comprises of hierarchically arranged modular components operating at different time scales: one component that is fast and invariable, and another that is slow but variable. The computational framework used here can formally describe such neuroethological processes.

## Methods

**Breeding moths for experiments.** The experiments described in this paper were performed on 1–2-day-old adult Oleander hawkmoths, *D. nerii*. The moths were either laboratory-cultured or reared from wild-caught pupae. Moth eggs from the laboratory culture were obtained by placing two male and two female moths in a large meshed chamber (~8 m³) along with their host plants, *Nerium oleander* and *Tabernaemontana divaricata*. Additional flowering plants placed in the meshed chamber provided nectar for the adult moths. Moth larvae were placed in mesh-topped boxes and reared on a natural diet of *N. oleander* leaves. Post-pupation, they were placed in sawdust until the adult moths emerged. The emerged adults were placed in soft cloth cages and exposed to natural day–night cycles until they were used for experiments.

**Delivering airflow stimulus.** A customized and calibrated wind-tunnel was used to provide moths with different airflow cues. The working section of the wind-tunnel had a cross-section of 0.28 m × 0.28 m and a length of 1.2 m (Fig. 1a). The floor and the walls of the working section were covered with white paper to minimize visual cues. Moths were tethered such that the head of the moth lay approximately at the centre of the working section. The airspeed in the wind-tunnel was monitored using a constant temperature mini-anemometer (Kurz 490S, Kurz Instruments Inc., Monterey, CA, USA).

**Tethering via a dorsal magnetic tether.** Moths were first cold anesthetized by placing them in −20 ºC for 7–9 min. The anesthetized moths were placed on an

aluminium block maintained at 0 °C, their dorsal thorax was descaled and a neodymium magnet (3 mm diameter, 1.5 mm thick) was attached to the area (Supplementary Fig. 1A). The total duration of the tethering process, including the cold anaesthesia, did not exceed 20 min. For the antennal perturbation experiments (described below), a small piece of a minutien pin (Austerlitz insect pins) was glued to the tip of the left antenna using UV-cure glue (Loctite-352). Moths were allowed to recover from the process for at least an hour. After recovery, moths were tethered using the dorsally glued magnet and placed in the wind-tunnel. This method of tethering is similar to that in Hinterwirth and Daniel (2010)[52].

**Restricting JO**. Moths were cold-anesthetized and placed on an aluminium block maintained at 0 °C throughout the procedure. The area around the base of the antenna was carefully descaled using a fine brush to ensure that the Böhm's bristles at the base of the antenna remained intact during the descaling (Supplementary Fig. 1B, C, E). Next, the pedicel–flagellum joint was descaled carefully under the microscope. A small drop of cyanoacrylate glue was placed on the pedicel–flagellum joint and spread around the joint using an insect minutien pin (Austerlitz insect pins) (Fig. 1c; Supplementary Fig. 1F). For sham-treated moths, the glue was placed on the third/fourth annulus of the antenna instead of the pedicel–flagellum joint (Fig. 1c; Supplementary Fig. 1D). After both the antennae were treated, a neodymium magnet was glued to the thorax of the moth (described above). The entire duration of this procedure for both the sham and the JO-restricted moths was around 30–40 min.

**Antennal response to airflow**. Tethered moths (control, sham and JO-restricted) were presented with frontal airflow ranging from 0 to 5 m s$^{-1}$ in increments of 0.5 m s$^{-1}$. The antennal response to airflow was filmed at 100 fps using two Phantom v7.3 high-speed cameras (Vision Research, Wayne, NJ, USA; Fig. 1a). To ease digitization, a black spot was marked at ~5 mm from the tip of each antenna for all treatments (control, sham and JO-restricted). We recorded at 100 frames for each wind-speed, which corresponds to about 35 wingbeats of antennal position data per wind-speed. Most moths initiated flight immediately after tethering. In the few cases when flight was not automatically initiated, we elicited flight either by giving them a brief airflow of 0.5–1 m s$^{-1}$ or by a tactile stimulus to the abdomen. Once a flight bout was initiated, the moth was given increasing airflow from 0 to 5 m s$^{-1}$ and the antennal response was continuously recorded. In general, antennal response was obtained in one continuous flight bout of the moth. After every experiment, the filming area was calibrated using a custom-made 3D calibration object. The recorded videos were sub-divided into antennal responses for individual airflow speeds, then calibrated, digitized and analysed.

**Antennal perturbation at different airflows**. In this experiment, tethered moths were given four values of airspeed: 0, 1.5, 2.5 and 4 m s$^{-1}$. These values were chosen such that they represented the entire curve obtained from the above experiment (Fig. 1d). Two custom electromagnets were used to perturb the antenna of either control or JO-restricted moths, at the above four airflow speeds. The electromagnets were positioned above and behind the tethered moth such that, when switched on, they pulled the antenna backward and slightly upward. Only the left electromagnet was used for perturbation, whereas the one on the right was placed symmetrically to avoid differential visual inputs to the moth. The antennal responses to perturbations at different airflows were recorded at 1000 fps using two Phantom v7.3 high-speed cameras (Vision Research, Wayne, NJ, USA). An Arduino mega 2560 was used to switch on and off the electromagnet five times per trial, at a frequency of 1 Hz. The perturbation protocols were flanked by quiet zones to record the resting position of the antennae. The total duration of perturbation for each airflow was ~7 s. To ensure consistent ratios of quiet zones and perturbations, the camera was triggered by the Arduino before beginning a protocol. The data was usually obtained from one flight bout for each protocol.

**Data analysis—antennal response to airflow**. Almost all the data analysis and statistics used in this paper were done in MATLAB (The MathWorks, Natick, MA, USA). Regression fits, on the other hand, were all performed in R (R Core Team, 2017)[53].

Using the two different calibrated camera views, we reconstructed the three-dimensional Cartesian coordinates of the tip and the base of both the antenna. Digitization of the antenna and reconstruction of the digitized points were done in MATLAB using a custom written code (DLTdv5[54]). Antennal vectors were calculated using the Cartesian coordinates of the tip and base of each antenna. The angle between the antennae, also called IAA, was computed as the angle between the two antenna vectors. All 100 frames of antennal position for all airflows were first digitized for a subset of the data (2 out of 8 control moths, 2 out of 5 sham moths and 5 out of 9 JO glued moths). The mean and the standard deviation of IAA for each of these moths were then compared to 10 randomly picked frames from the same dataset. The mean and standard deviation of just 10 digitized frames were comparable to the mean and the standard deviation of all digitized frames. Therefore, for the rest of the dataset, only 10 frames were digitized out of the 100 recorded frames.

After digitization, the IAA data was imported into R. Spearman's coefficient was computed for control, sham and JO-restricted moths' responses to airflow. The

coefficients for all moths were then pooled together based on treatment and tested for normality using Lilliefors test. Because the coefficients for all three treatments were not normally distributed, Kruskal–Wallis and Nemenyi test were used to detect statistically significant differences between the three treatments.

The sensitivity of antennal movements to airflow was computed as change in interantennal angle per 0.5 m s$^{-1}$ step change in airflow (Supplementary Fig. 2D–F). The torque on the antennal base due to aerodynamic drag was non-linear and dependent on the antennal angle and speed of frontal airflow[30]. It was computed using the following equation (derived from ref. [30]):

$$T_\gamma \sim \sin \gamma . v_a^{1.44} \qquad (1)$$

in which $\gamma$ is the antennal angle and $v_a$ is the speed of airflow. $T_\gamma$ was calculated for all three treatments (using data in Fig. 1d–f). If feedback from JO regulates torques by changing antennal position, $T_\gamma$ should remain constant for increasing airflows. Airflow-dependent changes in antennal position mitigated $T_\gamma$ (Supplementary Fig. 2G–I; compare control, sham with JO-restricted), but did not maintain it at a constant value.

**Data analysis—antennal perturbation at different airflows**. Antennal responses to perturbations were recorded at 1000 fps by two high-speed cameras for ~7 s per airflow, for four different airflows. Because of this high frame rate, the number of frames to be digitized per moth was 28,000 per camera view. The partially manual method of digitizing used for the above experiment was not feasible here. Therefore, an automated method of digitization was developed and used for this experiment. A custom code, henceforth called Autotracker, was written using the computer vision toolbox in MATLAB to automatically track the moth antennae and the tip of the moth's head. Autotracker requires only the first frame (and a few problematic frames) to be manually annotated. It uses the annotated frames as templates and tracks the antennae by using a combination of template-matching score maximization, point drift minimization and error residual minimization (https://github.com/AbstractGeek/Score-Based-Autotracker). At every frame, manually annotated templates were compared with sub-images extracted from the image. A template-matching score was computed as the distance between individual pixels in the sub-images and the templates. To minimize point drift to similar structures in the vicinity (from tips of antennae to tips of legs, for instance), we assumed a maximum velocity of the antennal movements. This method also restricted the search space of the template-matching algorithm, thereby increasing its speed.

By matching templates, we obtained a distribution of similarity, ideally centred around the point which was being tracked, for both the camera views. Because the actual tracked point was an object in 3D, the tracked points in both the camera views should correspond to the same point in 3D. This occasionally failed if we used the highest matched template in both the views, because of small variations in lighting which moved the position of the highest matched template in one view, but not the other. To circumvent this problem, we used the top 50 matched templates from both the views to calculate the error residual, i.e. the 3D reconstruction error[54]. We found a 3D point which minimized the 3D reconstruction error while simultaneously maximizing the similarity of templates in both the views. The 3D reconstruction code was adapted from Hedrick (2008), and DLTdv5 was used to validate all computations[54]. Using this approach, we were able to robustly track the points of interest in our videos and reconstruct them in 3D.

We used Autotracker to obtain the 3D Cartesian coordinates of the tip and the base of both antennae, and the tip of the moth's head. The output of the autotracker was checked for mistracked frames and manually corrected if found. The mistracked frame correction was performed using a modified version of DLTdv5, which uses the same template-matching algorithm as the Autotracker (https://github.com/AbstractGeek/Score-Based-Autotracker/tree/master/Improved-DLTdv5)[54].

Using Autotracker, we first completely digitized a subset of the dataset (4 out of 11 control moth trials and 2 out of 6 JO-restricted moth trials, Supplementary Fig. 6A, B). For the rest, we completely digitized only the regions where the electromagnets were switched off and the antennae were free to respond, as digitization of the whole dataset was not necessary (Supplementary Fig. 6C, D). For instance, when the electromagnet was on, the antenna and head positions remained relatively constant. Hence, using only a few frames before the antenna release was sufficient for obtaining the perturbed location of the antenna. Additionally, the distributions of the digitized points obtained from this dataset were comparable to the distributions from the complete dataset.

After correcting for tracking errors, the Cartesian coordinates of the digitized points were used to compute the antenna vectors and the head vectors. Antennal vectors, like above, were computed from the Cartesian coordinates of the antennal tip and the base points. The head vector, on the other hand, was computed as the unit vector along the line connecting the midpoint of the antennal bases to the head point. Individual antennal angle was defined as the angle between an antenna vector and the head vector. Because the head is free to rotate in our experiments, a head-centric definition allowed us to compute the antennal angle independent of head rotation (Supplementary Fig. 5A–D). During electromagnetic perturbations, we observed small changes in head orientation throughout our experiments (Supplementary Fig. 5E-G). Despite this, the average position of the right (control) antennal angle typically remained constant throughout the trial, suggesting that the

head-centric method used to compute antennal angles eliminated rotations of the head (Supplementary Fig. 5A–D, Supplementary Fig. 6). Additionally, both active head rotations and the ones elicited by electromagnetic perturbation seem to be too small to affect airflow-dependent antennal positioning (Supplementary Fig. 5A–D).

Antennal set-point was defined as the angle the antenna was corrected to after each perturbation (steady-state angle after perturbation). Set-points were statistically compared using the same methodology as for the IAA in the above experiment. After importing the corrected angle data set into R, Spearman's correlation coefficient was computed for set-point vs. airflow for every moth (control/JO). The coefficient for all moths were then pooled together based on treatment and tested for normality using Lilliefors test. Because the coefficients were not normally distributed, Wilcoxon rank-sum test was used to detect statistically significant differences between the two treatments.

Next, the antennal return trajectory was extracted from the left (perturbed) antennal angle. The return trajectory, however, was influenced by the wingbeat (Supplementary Fig. 5H), likely due to transmission of wing motion via the head to the antennae and due to induced airflow produced by the beating wings[2,55]. Such wingbeat-induced noise is typically removed using a simple notch filter about the undesired frequency. However, this approach would have filtered nearby frequencies and/or added ringing effects, thereby modifying the characteristics of the return trajectory. We circumvented this problem by subtracting the wingbeat frequency from the return trajectory. To do this, we first obtained the Fourier transform of the first derivative of the antennal trajectory, where the wingbeat frequency was most prominent. Generally, we found two dominant frequencies in the wingbeat frequency range (35–40 Hz). These two frequencies might arise due to slight drifts in wingbeat frequencies due to the long recording duration (the separation between these frequencies were generally around 1 Hz). These two dominant wingbeat frequencies, along with their first harmonics, were selected and used to fit sine waves (amplitude and phase were the two free parameters for the fitting algorithm). These were then subtracted from the first derivative of the antennal return trajectory, and the resultant was integrated to obtain the wingbeat frequency-free return trajectory. To validate this approach, we compared the effect of notch filtering vs. sine subtraction on raw data (Supplementary Fig. 5H). We used a fourth-order notch filter with a bandwidth of 5 Hz around the wingbeat frequency, which was determined as the highest power sinusoid with frequency from 35 to 45 Hz (wingbeat frequency of moths). The sine subtraction method produced very similar trajectories to the notch filtered one even for antennal trajectories with greater wingbeat-induced noise, which illustrates its effectiveness (Supplementary Fig. 5H). Sine subtraction ensured that only a particular frequency of a specific phase and amplitude was removed, ensuring that the characteristics of the return trajectory remain more or less unaltered.

After subtracting the wingbeat frequency from the antennal kinematics, we calculated antennal set-points from the return trajectories and used these as inputs to tune the control theoretic models. After the electromagnetic perturbation was turned off, the antenna initiated its return to its set-point. The switching off of the electromagnetic field itself, as measured using a Hall effect sensor (DRV5053), was not a precise step function and took about ~75 ms to reach zero (Supplementary Fig. 5J, K). Hence, the point at which the electromagnet stopped did not provide an accurate indication of the end of perturbation. The onset of antennal return movement was inherently variable from trial to trial because it was dependent on many factors including the distance of the electromagnet, the differences in antennal inertia, etc. (Supplementary Fig. 5H, I). Instead, we determined the angular difference between the angle at which the antenna was held by the electromagnet to the angle at which it finally settled (set-point), and arbitrarily defined 25% of this difference as the start of the return trajectory (Supplementary Fig. 5H). The set-point, the start point and the return trajectories were stored for further analysis (control theoretic model described below).

**Control theory models of antennal response to perturbation.** To analyse the return characteristics of the antennae, we modelled the antennal circuit as a closed-loop feedback system with set-point as the input and the measured antennal position as the output (Fig. 4a). The transfer function of the complete system depends on the transfer function of the antennal circuit $[L(s)]$. We systematically increased the number of poles (np) and zeros (nz) in $L(s)$ from zero to two (a similar approach to ref. [35]). However, we were unable to incorporate delays into our transfer functions due to inherent variable delay in electromagnet release. Below is a list of transfer functions (system models), along with the number of poles and zeros per system (Fig. 4a).

1. [np = 0, nz = 0] Proportional (P) system
2. [np = 1, nz = 0] Integral (I) system
3. [np = 1, nz = 1] Proportional Integral (PI) system
4. [np = 1, nz = 1] Proportional Differential (PD) system
5. [np = 2, nz = 0] Double integral (II) system
6. [np = 2, nz = 2] Proportional Differential Integral (PID) system

We next described the transfer function for each of these models (Fig. 4a), converted it into a state space model with the constants as free parameters, set-point as the input and the antennal return trajectory after release as the output. By doing this, we were able to estimate the constants of the complete antennal-positioning reflex for input set-points and output antennal angles. We found the

best parameter fits for each model for the given input and output using the System Identification Toolbox in MATLAB. The coefficient of determination ($R^2$) and normalized Akaike information criterion (nAIC) were computed for each of these models to estimate the goodness-of-fit (Fig. 4e). The coefficient of determination ($R^2$) quantifies the amount of variation in the raw data explained by the model. The normalized Akaike Information Criteria (nAIC), on the other hand, quantifies how much information is lost if one uses the model instead of the raw data. Both estimates, when combined, provide a measure of the goodness-of-fit for each model. Additionally, both these coefficients were statistically compared across models using Kruskal–Wallis and Nemenyi test (not normally distributed, Lilliefors test, $p < 0.001$).

To test the predictive capabilities of all the models, we computed the model constants based on the measurements from one single airflow and used it to predict the return trajectories of the antennae for another airflow. We performed this analysis repeatedly for all airflows and computed the $R^2$ coefficients and the nAIC values for all the fits. These coefficients were also not normally distributed (Lilliefors test, $p < 0.001$). We, therefore, compared them statistically using Kruskal–Wallis and Nemenyi test.

**Spiking neural circuit model of the antennal circuit.** The parameters of the components of the minimal neural circuit were set based on the integral (I) system model. Overall, we kept the spiking neural circuit model extremely simple, with the number of assumptions at a bare minimum, and used it as a feasibility test to check if such a neural circuit can produce behaviour similar to that seen in experiments.

The minimal spiking neural circuit model incorporated the equivalent components of a simple linear integral model and had four components:

1. Mechanosensory neurons of the Böhm's bristles were assumed to be simple on-off neurons which encoded antennal position at the population level. Firing rates of individual neurons were dependent on activation (On: 50 Hz Poisson firing, Off: 10 Hz). Recruitment of the sensory neurons was proportional to activation of the hair plates (Böhm's bristles), i.e. the antennal angle.
2. Interneurons modulated the set-point of individual motor neurons based on activity of JO, which in turn depended on airflow. To break symmetry and cause the antenna to move forward with increasing airflow, the interneuron activated one of the motor neurons preferentially more than the other (Fig. 5a).
3. The motor neuron was a simple integrate-and-fire neuron whose output firing rate was proportional to synaptic input. Thus, it effectively acted as an activity summator in the control theoretic model (Fig. 4a). The negative feedback arises from the connectivity between motor neurons and the muscles they activate; motor neurons activate muscles which, upon contraction, reduce the mechanosensory feedback from the hair plates, in turn decreasing their own activity (Fig. 5a).
4. Muscles, due to their slow calcium dynamics, integrate the error signal (the time constant of the integral system is in the same range as calcium integration times[4,38]). The combined activity of all antennal muscles determine change in antennal position. For the simple scape–pedicel joint, changes in position depend on the difference in activity between the two muscles (Fig. 5a) and reach equilibrium when the activities are equal.

The sensory neurons model the mechanosensory neurons that innervate hair plates (Böhm's bristles). The activation of the hair plates is binary - the mechanosensory hairs are either active (bent) or inactive[14]. In cockroaches, activation of these sensory neurons has a phaso-tonic response to bending, with the phasic component depending on the velocity of bending and the tonic component on the degree of bending[56,57]. A phaso-tonic firing response necessitates including velocity into the calculations, making both the sensory neurons and muscle models parameter heavy. To simplify the circuit and reduce the number of parameter assumptions in the model, we excluded the phasic part of the response and fixed the tonic firing rate to 50 Hz. The sensory neurons in the model, therefore, change their firing rate from a resting firing rate of 10 Hz to 50 Hz (Poisson) after activation.

The interneuron in the model was an excitatory neuron whose firing rate changed with airflow (set-point). The motor neuron was approximated by an integrate-and-fire neuron which received inputs from sensory neurons and the interneuron (generic integrate-and-fire parameters[58]). Every spike from the motor neuron elicited a calcium spike in the muscle, which decayed exponentially with a time constant of 50 ms (based on the integral system from the control theoretic analysis). The contraction of the muscle, at every time step, was determined by the level of calcium in the muscle. Two such circuits were used to model the pedicel segment of the antenna (Fig. 5a). The time step for the neural circuit and the associated calcium dynamics of the muscles was 0.1 ms. The change in antennal position, which was determined from the contraction of the antennal muscles, was updated every 10 ms. The model was simulated using NEST[58] and the antennal position was analysed in MATLAB. To compare the performance of the neural circuit with behavioural data, we ran the simulated data through the same control theoretic analysis. Because the sampling rate was 1000 fps for the model, the simulated position was upsampled using a cubic spline interpolation. The simulated position was then run through the same control theoretic framework

described above. The exact same analysis, including statistics, was also used on the simulated data.

**Reporting summary**. Further information on research design is available in the Nature Research Reporting Summary linked to this article.

## Data availability
The data that support the findings of this study are available in Zenodo with a unique identifier of https://doi.org/10.5281/zenodo.3515753. Raw experiment videos from which the data was digitized are available upon request from the corresponding author.

## Code availability
Analysis codes are available in Github and Zenodo with a unique identifier of https://doi.org/10.5281/zenodo.3515775. Autotracker codes used for digitization are available in Github and Zenodo with an identifier of https://doi.org/10.5281/zenodo.3517748.

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

## Acknowledgements

Funding for this study was provided by grants from Air Force Office of Scientific Research (AFOSR) # FA2386-11-1-4057 & # FA9550-16-1-0155, & National Centre for Biological Sciences (Tata Institute of Fundamental Research) to S.P.S. D.N. was supported by EuroSPIN Erasmus Mundus Joint Doctoral Programme.

## Author contributions

D.N., N.S. and S.P.S. designed the experiments. D.N. and N.S. performed the experiments, digitized and analysed the data. D.N. wrote all the necessary software to analyse the data. D.N. and Ö.E. designed the simulation experiments. D.N. performed the simulations and analysed the data. D.N., N.S., Ö.E. and S.P.S. wrote the paper. S.P.S. and Ö.E. supervised and obtained funding for the project.

## Competing interests

The authors declare no competing interests.
