## [Peer Review File · Nature Communications]

Reviewers' Comments:

Reviewer #1:

Remarks to the Author:

The authors use a combination of restriction experiments to test the necessity of the Johnston's Organ and modeling of the faster Boehm's Bristle reflex to pose a hierarchical control scheme for the antennae positioning of hawkmoths in different windspeeds. The magnetic perturbation of antennae and role of the bristles in mediating the response was done in their previous paper. They have certainly gone beyond that here by showing that the set-point can be modulated by an outer-loop control from the JOs and that the bristle reflex can be modeled with an integral control. While this manuscript takes the next step in advancing this story, I have a number of specific concerns. Overall it seems to take a solid step towards integration but My other comments are about the fitting and modeling which I think are important but potentially addressable and shouldn't take away from nice experimental work.

1. I think the authors may need to make more clear what the significant impact of this paper is in light of their previous work. The bristle feedback was already established in their nice recent paper and the JOs are known to be involved in positioning. Here the real advance is putting these pieces together by showing that the JOs adjust the set point and capturing the bristle response with an I controller. I worry that this framing takes the story one step further but does yet fully integrate the two time scales in a way that might translate for those beyond field specialists. The modeling seems to capture the bristle response while the JOs adjust the system but aren't included dynamically. I do very much like that the model fits each of the steady windspeed conditions, but could the JOs be explicitly brought into the control model? This would seem to strengthen the exciting prospect of two neural circuits acting at different time scales regulating this behavior in an integrated control loop.

2. I had some confusion about how the bristle response could be fast, but use a slow controller. This seems at odds with the introduction some of the interpretation. The author's make a point that simple summation by the neural controller would be fast, but that seems a bit odd because if the mechanics are a pure I controller that is typically very slow. This is seen in recovery traces where it takes a couple hundred milliseconds for the position to recover. Can the authors reconcile the longer time course of the behavior and the apparently slow mechanical response of the muscles & mechanics with the time course separation between bristles and JOs? My understanding was that the JOs would have started reacting in the time course they are suggesting for the inner loop, bristle mediated response. If this is challenging it might further suggest that the delays really should be incorporated into the model.

3. Along these lines most neural systems have real and significant delays. Does the incorporation of a delay term into the transfer function substantially change the fit even for just the bristle part of the control loop? See for example Madhav et al. JEB 2013 which fit variable zero and pole transfer functions with and without delays.

4. The authors have a quick statement about PI being like PD but with band limits. This doesn't quite translate into the mechanistic interpretation though. A PI on a positional error suggests a very different circuit and responsiveness to a PD on positional error. Being more explicit in this and directly considering a PD model might help. PD models are common in the literature (as are PI) and when talking about fast behaviors band limits might matter.

5. It was unclear in my reading whether the authors were treating the neural network model as a biophysical or phenomenological/predictive model. I have some concerns in either case mostly that a high parameter neural network can frequently predict the output of a system without much connection

to the underlying circuit. From the mechanistic modeling standpoint it seems more interesting how this behavior could arise from realistic neural firing and muscle models. I'm worried that the binary firing rate and extremely simplified muscle model means we cannot treat this as a circuit level model to guide experiments or model iteration. From a phenomenological modeling standpoint the analysis of the I controller model seems to do just as good a job as the much higher parameter neural network model.

I think the authors meant the neural network model as somewhere in between, maybe as a feasibility test? The motivation for doing this model and how it integrates with the other results should be strengthened.

6. Can a single type of disturbance to the antenna really well distinguish between the different controller models? On line 321 the authors report a large range of responses but they step responses all seem pretty similar especially in time course to me. If other amplitudes or a varying disturbance were played in might that better distinguish the potential importance of P and D terms? Since the fits are not perfect, do we need to consider non-linear models?

7. How close were the magnetic perturbations to a step function? I imagine they were fast but want to make sure we aren't looking at the response of electromagnet.

8. Is head rotation during the unilateral perturbations or their recovery significant? This seems like a very important control for the interpretation of the data.

Line 494-502 The vague statement that "more details will be published elsewhere" is a bit concerning here. I suspect the program works quite well but there is a lot in this code that cannot be easily vetted. The original DLTdv5 should probably be cited here too.

Line 504 the "interesting subsections" is a bit vague. I assume from the following that you mean the periods immediately around the perturbations? Can you specify?

Line 534-543 I am a little concerned about the subtraction of fitted sinusoids here, especially because the wingbeat frequency doesn't have a single sharp peak. I agree that this suggests that wingbeat frequency might be being slightly modulated during the trials. The trouble is that this would create a time varying response. Subtracting out fitted sinusoids implies that all those components are present throughout the duration of the trial. Normally I wouldn't be too concerned but if a notch filter (which is very similar) does not work I am concerned about these methods too. I understand that the authors may be stuck with a tricky data analysis point here. My recommendation is show a (supplementary) figure with the raw and sin subtracted data and to more carefully illustrate the effect of what was done here. At least then the reader can assess the effect.

Lines 126 The statement that all cases were monotonic (not "monotonously") is not strictly true (there are some small rises) but is generally true (significant spearman correlation).

Lines 557-579 The methods here are very similar to those in Madhav, et al. 2013 JEB. This paper should probably be cited and if there are differences they should be discussed.

L 599-600 it's good to compare to the cockroach hair plate work but if those neurons are typically phaso-tonic what's the basis for thinking of them as purely on-off?

L645 s.e.m. is usually standard error of the mean. Did the authors mean that or margin of error?

L 692 convoluted → convolved?

L 707 I don't understand how the R^2 of the P controller can be negative. Also the numbers don't agree with the statement. The PID controller appears to have the highest R^2 (0.82) despite the statement say I is higher.

I found the italics a bit over-used and distracting.

Reviewer #2:

Remarks to the Author:

In this manuscript Natesan et al. explore the differential roles of two mechanosensory organs — Böhm's bristles (BB) and Johnston's organ (JO), in the control of antennal positioning. The work follows an earlier study from the same group (Krishnan et al. 2012) in which they established that BB, but not JO, are required for antennal positioning during flight, and suggest, due to short latency of activation, that BB sensory neurons may make direct synaptic contacts onto antennal motor neurons. The present study argues based on stabilization experiments that JO inputs determine the setpoint of a reflex mediated by BB. They fit the dynamics of antennal responses to mechanical perturbations with both a control theoretic model and a neural circuit model.

The questions raised by this study— how is input from different sensors combined to control a motor output— are of broad interest, and the system described here for studying them, in which two inputs can be isolated anatomically and the motor output studied directly, seems ideal for addressing these questions. The experiments that are performed appear to have been done carefully and rigorously. However, the broad conclusions of the paper as set out in the Introduction and Discussion are not fully supported by the experimental data presented. Only the JO are directly manipulated in this study, limiting the conclusions that can be drawn. In addition, while the modeling presents proof-of-concept that a simple neural circuit could give rise to the observations, it does not demonstrate that this is the circuit that actually underlies them, as no neural data are presented.

Major concerns:

1) The study is motivated by describing how antennal positioning is modulated by multimodal cues, including wind, optic flow, and visual motion, as well as slower state-dependent cues such as flight versus walking. However, the study only examines the role of a single mechanosensory cue — wind speed — in flying moths. The study would be greatly enhanced by examining at least one other cue (flight state, optic flow, etc.) to see whether the same principles apply. For example, can all inputs be described as changing the antennal setpoint? Does the antennal positioning reflex operate in the same way when the moth is not flying?

2) The study focusses on the interaction between BB and JO mediated signals, but only JO is manipulated in the experiments. The conclusions about the role of BB and the circuit underlying its action rest entirely on the previous paper (Krishnan et al. 2012). It would be helpful for the authors to clarify how the experiments described in that paper (Figure 2) differ from those shown in Figure 1 of the current manuscript. At least it seems like the authors should repeat the main experiments shown here (antennal response to different windspeeds and to perturbations at different wind speeds) in BB-stabilized moths to directly compare the roles of the two organs.

3) Two models are presented— a control theoretic model and a neural circuit model. In the case of the control theoretic model, different forms for the feedback term are explored, but different control architectures are not. That is, the model assumes that BB provides rapid feedback and JO alters the

setpoint of the feedback. The neural circuit model contains several details about the structure of the neural circuit involved but is not supported by any data. It is nice to see that the model works but this does not demonstrate that this and not some other circuit architecture underlies the observed behavior. I think the modeling would have a greater impact if the authors could use this to show that their interpretation— that JO is determining the setpoint of a rapid BB-mediated reflex— is the only one consistent with the data.

4) Several speculative statements in the Introduction and Discussion are presented without proper citations or data to back up these ideas. For example, lines 43-49 and 50-51 make multiple claims about the antennae and flight control without references.

Minor concerns:

It is not clear from the Results and Figure legends that all data are from flying flies. This should be made clear.

The data shown in Figure 2C-F and Supp. Fig. 2 D-G are key to the paper. It would be helpful to show both of these in main figures so reader can directly compare them. Currently the supplementary figures lack figure legends, making the colors difficult to interpret. These same traces show intriguing dynamics that are not discussed in the text. For example the yellow trace in Fig. 2E shows some slow dynamics in the return to baseline. The authors should comment on this and note how these kinds of dynamics impact the interpretation of their model.

line 69: please clarify and justify what is meant by "trade-off". Also in Discussion line 373.

lines 122 and 131: maybe these don't need to be separate section headings?

lines 152-153 "Ablation of the hair plates" This statement needs a reference.

lines 152-155 How does "stable antennal positioning" differ from "airflow-dependent antennal positioning." Do the Bohm's bristles play a role in one context, but not in another?

line 179/Figure 2: It might be easier for the reader if the data were presented first, followed by the conceptual model. In addition, the control theoretic model in Figure 3A could be arranged graphically in the same way as the conceptual model in Figure 2A.

line 190/Figure S2: The set point is not always the same in these data. Is this because the wind passively deflects the antenna to a different position? Please provide an explanation.

line 240: linear integral system. Is it clear that this integration time reflect neural control rather than passive mechanical properties of the antenna? What happens in a dead fly?

line 246: the neural circuit model assumes that the MN itself determines the setpoint but no data are presented to support this hypothesis. Could it not be set somewhere more centrally?

line 287: does airflow-dependent modulation of antennal position occur in non-flying flies? If not, what does this tell us about the neural control of this reflex?

line 297: can the setpoint of the antenna be altered by other means (visual, odor, flight state) if JO is ablated?

line 314/figure. 3C: could these curves be fit with a single exponential decay? What do these curves look like if they are overlain and how do these compare to a single exponential decay? Is the control theoretical model necessary to account for these or could passive mechanical properties play a role?

353-354 I believe that the DUM innervates antennal muscles directly in some insects as well (Braunig et al. 1990, Allgauer & Honegger 1993).

Reviewer #3:

Remarks to the Author:

This paper examines the feedback control system that underlies how moths regulate antennal angle with respect to airflow during flight. While the scope of the paper is narrowly focused, the studied reflex has important behavioral implications to flight stability, state-dependent sensory tuning, and multi-sensory integration. The main result of this work delineates the contributions of two separate antennal mechanosensory structures: the Johnston's organ (JO) and Bohm's bristles. The JO mediates antennal repositioning in response to changes in air flow; Bohm's bristles provide the fast sensory cues that help regulate the antenna to the set-point angle. The authors present a thorough and convincing analysis, incorporating elegant tethered-flight perturbation experiments, a comparison of control theoretic models fit to data, and a proposed neuronal instantiation of the model. Such models may also inform understanding of other mechanosensory-motor feedback loops in insects, such as those that regulate wing or leg kinematics.

Additionally, the manuscript is thorough and well-written and the figures are clear and informative. I have no major concerns regarding this manuscript, so below I outline my moderate and minor comments.

Moderate comments:

Regarding airflow, antennal position, and drag force:

You report that for a given airflow, moths exhibited a distribution of set angles (Lines 125-126 and Figure 1 D, E, F). If we consider the antenna as a beam, its deflection would be proportional to drag force. But at different antennal positions, the wetted area will be proportional to $\sin(\theta)$ and the aerodynamic drag on the antenna would have a $(u \cdot \sin(\theta))^2$ term.

- Each antennal position changes the sensitivity of the sensor, airflow/force or airflow/deflection depending on what quantity the JO is measuring. Might moths be regulating the sensitivity of their airflow sensor, with individuals having different preferred sensitivities? Rather than plotting change in antennal position vs airflow, it might be interesting/informative to plot sensitivity (raw or baseline subtracted) vs airflow.
- At least, you should show a distribution of antennal baselines at 0 m/s. From Fig 1D, some moths exhibit nearly 90 deg motion towards the center at the highest air flows! Does this suggest that the inter-antennal angle would be 180 deg (at least)?
- In the Methods section (Lines 515-517), you mention that the head is free to move. Head movements would also affect the relative angle of the antenna with respect to air flow. Could you provide a distribution or approximate range of head movement? Are head movements significant enough to influence the proposed model?

Regarding parametrized control theoretic models:

Please report the best-fit parameters for the different model structures. In Figures 3B and 4D, the PI and I models coincidentally yield fits with identical goodness (down to the 0.01%). Is the P-value

negligible? That is, did the proportional-integral model in effect reduce to the integral model? If so, that is an even stronger case for the I-model than just parsimony.

Minor edits

Line 45: "For optimal acquisition of sensory information and robust flight control..."

Firstly, we might suspect that's why insects move their antennae, but I don't know of evidence of this in the literature (if there is, it should be cited). Secondly, in the context of feedback control (mentioned in the second clause), "optimal" and "robust" have specific meaning. This paper does not evaluate optimality or robustness. I would weaken this clause, perhaps:

Insects actively move and position their antennae and eyes via fine feedback control of antennal muscles and head/eye movement, putatively to tune sensory sensitivities and stabilize flight.

Line 53, 63 382: again, "optimal" and "optimizes" is over-reaching.

Line 161 (typo): "comprises of two sub-circuits" delete "of"

Line 181-183: "Thus, ..." This sentence seems like an overly wordy way of saying that the antennal angle changed due to airflow. What is the use of explicitly enumerating the motor neurons, motor commands, and antennal muscles that effect the change?

Line 200: "control theory model" should be "control theoretic model" or just "control model"

Optimal and robust

Neuronal network or neural circuit, so as to disambiguate from artificial neural network

Line 292, 323, 325, 329, and elsewhere: You describe your model as a neural network which sets up expectations of an artificial neural network. To avoid confusion with the oh-so-popular machine learning tool, I'd substitute either "neuronal network" or "neural circuit". Also, your model comprises very few cells; I think "circuit" is more accurately descriptive than "network".

Line 298: In referring to JO-inhibited preparations, "but cannot alter the set point." Cannot or do not? JO-inhibited moths do not alter the set point in response to air flow, but they might still alter the set point if presented with an optic flow stimulus, no? So they physically could, but they do not because of the sensory inhibition.

Line 383: delete "such".

Response to Reviewers' comments for NCOMMS-18-38698:

We wish to thank all three reviewers for their thorough reading of the manuscript, and for their very pertinent and thoughtful comments on our manuscript. We have tried to address their comments as comprehensively as we could, and feel that the manuscript reads so much better as a result.

Reviewer #1 (Remarks to the Author):

The authors use a combination of restriction experiments to test the necessity of the Johnston's Organ and modeling of the faster Boehm's Bristle reflex to pose a hierarchical control scheme for the antennae positioning of hawkmoths in different windspeeds. The magnetic perturbation of antennae and role of the bristles in mediating the response was done in their previous paper. They have certainly gone beyond that here by showing that the set-point can be modulated by an outer-loop control from the JOs and that the bristle reflex can be modeled with an integral control. While this manuscript takes the next step in advancing this story, I have a number of specific concerns. Overall it seems to take a solid step towards integration but My other comments are about the fitting and modeling which I think are important but potentially addressable and shouldn't take away from nice experimental work.

We thank the referee for a careful and thoughtful reading of our manuscript, and also for framing the comments in the context of our previous research, which led to this paper.

1. I think the authors may need to make more clear what the significant impact of this paper is in light of their previous work.

The key "take-home message" of the paper for non-specialists is the following: Fast behaviours in insects are often considered reflexive, and in neural terms sparsely connected and hard-wired. From a controls standpoint, such reflexes are formally evaluated as negative-feedback loops. Whereas a reflex circuit ensures rapidity of responses, it precludes variability that is often required to fine-tune the response with respect to the context. Such fine-tuning in response to additional inputs such as olfaction, vision, etc. can be slower, and controlled by accessory feedback loops. *So, any complex behaviour may be thought to contain a component that is fast and invariable, and another that is slow but variable.* This is the important general insight gained from the work described here. We have added this as a summary paragraph at the end of the paper, so we can leave the readers with this take-home summary.

1 (continued). The bristle feedback was already established in their nice recent paper and the JOs are known to be involved in positioning. Here the real advance is putting these pieces together by showing that the JOs adjust the set point and capturing the bristle response with an I controller. I worry that this framing takes the story one step further but does yet fully

integrate the two time scales in a way that might translate for those beyond field specialists. The modeling seems to capture the bristle response while the JOs adjust the system but aren't included dynamically. I do very much like that the model fits each of the steady windspeed conditions, but could the JOs be explicitly brought into the control model? This would seem to strengthen the exciting prospect of two neural circuits acting at different time scales regulating this behavior in an integrated control loop.

Our paper investigates the case example of JO- and Böhm's bristles-mediated feedback loops, by formally analysing them as hierarchically-nested control loops. Here, Bohm's bristles are most well-investigated. At present, the modulatory influence of JOs is somewhat more nebulous, because many details still need to be investigated, but we reckon that it will take several years to fully characterize JO-to-motor neuron circuits.

What needs to be achieved for this? First, we must neuroanatomically characterize which subsets of the JO scolopidia (which are range-fractionated) are specifically involved in detecting airflow and mediating set-point modulation. For this we need to establish a clear tonotopic map, which we have not yet determined. Second, another study from our lab (Sant and Sane, J Comp Neurol., 2019) shows that JOs do not seem to directly arborize on the antennal motor neurons, which means that an interneuron or more may be involved, raising the possibility of additional modulation – for instance, control of antagonistic muscles, as indicated in Fig 5A. Third, at the behavioural level, we need additional experiments to establish the latencies of response relative to the JO which could also add some insights to this circuit. Without these details, we are unable to include JO more explicitly at the present time. Nevertheless, this is very much where this project is headed next.

2. I had some confusion about how the bristle response could be fast, but use a slow controller. This seems at odds with the introduction some of the interpretation. The authors make a point that simple summation by the neural controller would be fast, but that seems a bit odd because if the mechanics are a pure I controller that is typically very slow. This is seen in recovery traces where it takes a couple hundred milliseconds for the position to recover. Can the authors reconcile the longer time course of the behavior and the apparently slow mechanical response of the muscles & mechanics with the time course separation between bristles and JOs? My understanding was that the JOs would have started reacting in the time course they are suggesting for the inner loop, bristle mediated response. If this is challenging it might further suggest that the delays really should be incorporated into the model.

As may be expected, the rate limiting factor in the response is the time taken by the muscle to contract, and latency of transmission across the neuromuscular junction. As these are common factors in any antennal response irrespective of the precise controller dynamics, their dynamics governs the timescales of the antennal positioning response. Both JO and Bohm's bristles are mechanosensors, so their response is likely to be very rapid. However, JO response is (likely) routed through interneuron(s), whereas the

Bohm's bristles act directly on the motor neuron. We feel that the two timescales are likely separated by this intermediate circuit.

We are certain that the JO mediated response does not alter the return dynamics of the antennae for two reasons. First, gluing the JO does not alter the course of antennal return. Second, if JO response influenced antennal return, then we should have observed different antennal return dynamics as we altered airflow. However, we could predict antennal responses to different airflows using constants from just one airflow. This suggests that the influence of JO on these return dynamics may be very minimal.

3. Along these lines most neural systems have real and significant delays. Does the incorporation of a delay term into the transfer function substantially change the fit even for just the bristle part of the control loop? See for example Madhav et al. JEB 2013 which fit variable zero and pole transfer functions with and without delays.

In accordance with the referee's suggestion, we tried incorporating delays in the control model. However, there is considerable trial-to-trial variability in the antennal response which makes it difficult to fit any specific delay term (more details in #7).

4. The authors have a quick statement about PI being like PD but with band limits. This doesn't quite translate into the mechanistic interpretation though. A PI on a positional error suggests a very different circuit and responsiveness to a PD on positional error. Being more explicit in this and directly considering a PD model might help. PD models are common in the literature (as are PI) and when talking about fast behaviors band limits might matter.

Thanks for this suggestion. We agree and have redone the control theoretic analysis after adding the PD model. The PD model fits the data just as well as I, PI, and PID systems. We have added this to the figures (Fig. 4, 5, S3, S4), and the corresponding results and discussion.

5. It was unclear in my reading whether the authors were treating the neural network model as a biophysical or phenomenological/predictive model. I have some concerns in either case mostly that a high parameter neural network can frequently predict the output of a system without much connection to the underlying circuit. From the mechanistic modeling standpoint it seems more interesting how this behavior could arise from realistic neural firing and muscle models. I'm worried that the binary firing rate and extremely simplified muscle model means we cannot treat this as a circuit level model to guide experiments or model iteration. From a phenomenological modeling standpoint the analysis of the I controller model seems to do just as good a job as the much higher parameter neural network model. I think the authors meant the neural network model as somewhere in between, maybe as a feasibility test? The motivation for doing this model and how it integrates with the other results should be strengthened.

As the referee correctly surmises, our model is somewhere between a neural network model and a phenomenological (or empirical) one. It has empirical inputs and

predictive value, but within a mechanistic framework that was determined by our neuroanatomical and neurophysiological studies. As recommended, we have altered the text to elaborate upon the motivation for the neural network model and its relationship with other results.

Line 254-264: “The above experiments and control theoretic analyses show that airflow-dependent antennal positioning arises from an interplay between the antennal positioning reflex and a circuit that modulates the set point. Additionally, the antennal positioning reflex can be modelled as simple linear models which both fit and predict the antennal return dynamics irrespective of its set-point. Although these models provide a descriptive quantification of the computation underlying airflow-dependent antennal positioning, they do not provide a mechanistic hypothesis on how a neural circuit can perform these computations. Here, we proposed a minimal neural circuit that incorporates the simple linear models described above and simulated it as a feasibility test. Because a group of linear models (I, PI, PD, PID) fit and predict the antennal return dynamics equally well, we modelled the minimal neural circuit as an integral model, on the basis of parsimony (Fig. 4A).”

To strengthen the rationale for a model neural circuit, we have modified the text in the discussion.

Line 361-367: “Based on the linear models and the existing anatomical and physiological data, we proposed a minimal neural circuit model that could maintain and modulate position based on context. Because a whole group of models were able to fit and predict the error correction dynamics of the antennal positioning reflex, several neural circuits are possible. The integral model was used based on the parsimony of components and formed the basic framework for higher order models. We used this control theoretic approach to generate mechanistic hypotheses of underlying neural circuits that perform this computation.”

6. Can a single type of disturbance to the antenna really well distinguish between the different controller models? On line 321 the authors report a large range of responses but they step responses all seem pretty similar especially in time course to me. If other amplitudes or a varying disturbance were played in might that better distinguish the potential importance of P and D terms? Since the fits are not perfect, do we need to consider non-linear models?

We did consider non-linear models as our alternate hypothesis, but we did not find a reason to invoke them. Linear models both fit, and predicted, the time courses of antennal return trajectories to different amplitudes of step stimuli. That said, we agree that having other varying disturbance (like sum of sines, or white noise) might allow us to distinguish the importance of P and D terms. We have modified this section to add more clarity. This now reads:

Line 340-348: “The linear control theoretic models $L(s)$ captures the dependence of output position with respect to this error, with the integral model mathematically

equivalent to a decaying error exponential. Other higher-order models (PI, PD, PID) can be used to capture more complex dependencies between the output and the error. We provided different amplitudes of step perturbations, but this disturbance was insufficient to distinguish the difference in performances of the higher-order models. Other stimuli (e.g. sum of sines, white noise or chirps etc.) may likely provide better resolution for which of the four models (I, PI, PD, PID) best approximates the error correction dynamics of the antenna under a variety of disturbances [32–34].

Our alternate hypothesis was a non-linear model in which the error correction dynamics depend on the antennal set-point. Such dynamics could occur if modulatory inputs like optic flow or airflow alter not just the set-point, but also the time constants of the control theoretic models. Such a system can maintain the antenna at set point, but the error correction dynamics would change with changes in the set point. In such scenarios, predicting the antennal return dynamics at different set-points would not be possible based on just one set-point. To differentiate the above scenario from the linear model (Fig 2A), we quantified how well the models could predict the dynamics of the antenna in other airflows based on the dynamics of just one case. These predictions were only possible in the linear case, for which the underlying dynamics did not alter based on set point. The predictions explained a large range of return trajectories in all airflows (a median of 0.76 for I model, Fig S3), thereby suggesting that the linear model was sufficient.”

7. How close were the magnetic perturbations to a step function? I imagine they were fast but want to make sure we aren't looking at the response of electromagnet.

We have added a supplementary figure illustrating the on-off properties of the electromagnet and the various latencies associated with it. The electromagnets took ~110 milliseconds to switch on, and ~75 milliseconds to switch off completely (Fig. S5J-K). During this ~75 milliseconds, the magnetic field produced by the electromagnet gradually declined to zero (Fig. S5J). We expected the antenna to be released when the magnetic field falls below a certain threshold. We therefore independently computed the off-delay from the data based on when the antenna began to move. This had a distribution centred around ~50 ms, in both control and JO-restricted moths (Fig. S5I). This variability was due to differences in release thresholds between moths, which was dependent on many factors including the distance of the antenna from the electromagnet, the antennal inertia etc. This made the net delay in the release of the antenna more variable than the off-delay of the electromagnet (Fig. S5I-K). However, once released, it is very clear that the antenna is no longer affected by the decaying magnet field of the electromagnet, which is by then very close to zero.

We additionally modified the text to incorporate this detail.

Line 623-635: “After subtracting the wing-beat frequency from the antennal kinematics, we calculated antennal set points from the return trajectories and used these as inputs to tune the control theoretic models. After the electromagnetic perturbation is turned off, the antenna initiates its return to its set point. The switching off of the electromagnetic field itself, as measured using a hall-effect sensor (DRV5053), was not a precise step function and took about ~75 ms to reach zero (Fig S5J). Hence, the point at which the electromagnet stopped did not provide an accurate indication of the end of perturbation. The onset of antennal return movement was inherently variable from trial to trial because it was dependent on many factors including the distance of the electromagnet, the differences in antennal inertia, etc (Fig. S5H-I). Instead, we determined the angular difference between the angle at which the antenna was held by the electromagnet to the angle at which it finally settled (set point), and arbitrarily defined 25% of this difference as the start of the return trajectory (Fig. S5H). The set point, the start point and the return trajectory was stored for further analysis (control theoretic model described below).”

8. Is head rotation during the unilateral perturbations or their recovery significant? This seems like a very important control for the interpretation of the data.

To control for head rotations, we computed antennal angles relative to the head. We have added a supplementary figure illustrating head rotations and related description in the text.

Line 583-591: “Individual antennal angle was defined as the angle between an antenna vector and the head vector. Because the head is free to rotate in our experiments, such a head-centric definition allowed us to compute the antennal angle independent of head rotation (Fig S5A-D). Due to electromagnetic perturbations, we observed small changes in head orientation throughout our experiments (Fig. S5E-G). Despite this, the right (control) antennal angle typically remained constant throughout the trial, suggesting that the head-centric method used to compute antennal angles eliminated the rotations of the head (Fig. S5A-D).”

Line 494-502 The vague statement that “more details will be published elsewhere” is a bit concerning here. I suspect the program works quite well but there is a lot in this code that cannot be easily vetted. The original DLTdv5 should probably be cited here too.

We added more details to methods describe the algorithm underlying the code, with proper citations to the original DLTdv5. Thanks much for pointing out that we had failed to cite the original reference.

Line 543-564: “Autotracker requires only the first frame (and a few problematic frames) to be manually annotated. It uses the annotated frames as templates and tracks the antenna by using a combination of template matching score maximization, point drift minimization and error residual minimization (<https://github.com/AbstractGeek/Score-Based-Autotracker>). At every frame, manually annotated templates are compared with

sub-images extracted from the image. A template matching score is computed as the distance between individual pixels in the sub-images and the templates. To minimize point drift to similar structures in the vicinity (from tips of antenna to tips of legs for instance), we assume a maximum velocity of the antennal movements. This method also restricts the search space of the template matching algorithm, thereby increasing its speed.

By matching templates, we obtained a distribution of similarity, ideally centred around the point which is being tracked, for both the camera views. Because the actual tracked point was an object in 3-D, the tracked points in both the camera views should correspond to the same point in 3-D. This occasionally failed if we use the highest matched template in both the views, simply because small variations in lighting which can move the position of the highest matched template in one view, but not the other. To circumvent this problem, we used the top 50 matched templates from both the views to calculate the error residual, i.e. the 3D reconstruction error [47]. We found a 3D point which minimized the 3D reconstruction error while simultaneously maximizing the similarity of templates in both the views. The 3D reconstruction code was adapted from Hedrick (2008), and DLTdv5 was used to validate all computations [47]. Using this approach, we were able to robustly track the points of interest in our videos and reconstruct them in 3D.”

Line 504 the “*interesting subsections*” is a bit vague. I assume from the following that you mean the periods immediately around the perturbations? Can you specify?

Agreed – interesting subsections is a very subjective term, and best eliminated. We have replaced it with the following description to the text instead.

Line 572-574: “For the rest, we completely digitized only the regions where the electromagnets were switched off and the antenna were free to respond, as digitization of the whole dataset was not necessary.”

Line 534-543 I am a little concerned about the subtraction of fitted sinusoids here, especially because the wingbeat frequency doesn't have a single sharp peak. I agree that this suggests that wingbeat frequency might be being slightly modulated during the trials. The trouble is that this would create a time varying response. Subtracting out fitted sinusoids implies that all those components are present throughout the duration of the trial. Normally I wouldn't be too concerned but if a notch filter (which is very similar) does not work I am concerned about these methods too. I understand that the authors may be stuck with a tricky data analysis point here. My recommendation is show a (supplementary) figure with the raw and sin subtracted data and to more carefully illustrate the effect of what was done here. At least then the reader can assess the effect.

As recommended, we have added a supplementary figure (Fig. S5H) and updated the text to describe this comparison:

Lines 615-622: “To validate this approach, we compared the effect of notch filtering vs. sine subtraction on raw data (Fig S5H). We used a fourth-order notch filter with a bandwidth of 5 Hz around the wingbeat frequency, which was determined as the highest power sinusoid with frequency from 35-45 Hz (wingbeat frequency of moths). The sine subtraction method produced very similar trajectories to the notch filtered one even for antennal trajectories with greater wingbeat induced noise thereby illustrating its effectiveness (Fig S5H).”

Lines 126 The statement that all cases were monotonic (not “monotonously”) is not strictly true (there are some small rises) but is generally true (significant spearman correlation).

Agreed. We have modified this statement to better reflect the data. The new text reads as follows:

Line 126-128: “Although the initial inter-antennal angle was different for each individual, in all cases it generally decreased monotonically as airflow varied from 0 to 3 m/s, and stabilized when airflow exceeded about 3 m/s.”

Lines 557-579 The methods here are very similar to those in Madhav, et al. 2013 JEB. This paper should probably be cited and if there are differences they should be discussed.

Agreed. We have added the citation to the paper and slightly altered the text:

Line 641-645: “We systematically increased the number of poles (np) and zeros (nz) in L(s) from zero to two (a similar approach to [32]). However, we were unable to incorporate delays into our transfer functions due to inherent variable delay in electromagnet release.”

L 599-600 it’s good to compare to the cockroach hair plate work but if those neurons are typically phaso-tonic what’s the basis for thinking of them as purely on-off?

We have now added some text to elaborate why we assume those neurons to be purely on-off.

Line 700-707: “In cockroaches, activation of these sensory neurons has a phaso-tonic response to bending, with the phasic component depending on the velocity of bending and the tonic component on the degree of bending [52,53]. A phaso-tonic firing response necessitates including velocity into the calculations, making both the sensory neurons and muscle models parameter heavy. To simplify the circuit and reduce the number of parameter assumptions in the model, we excluded the phasic part of the response and fixed the tonic firing rate to 50 Hz. The sensory neurons in the model, therefore, change their firing rate from a resting firing rate of 10 Hz to 50 Hz (Poisson) after activation.”

L645 s.e.m. is usually standard error of the mean. Did the authors mean that or margin of error?

We did indeed intend to say “the standard error of the mean”. We have fixed this ambiguity throughout the manuscript as follows.

“The overlay around each line represent the standard error of the mean (s.e.m).”

L 692 convoluted → convolved?

Yes! We have fixed this mistake.

Line 823-824 “The error in position is convolved with a transfer function to obtain the output.”

L 707 I don’t understand how the R^2 of the P controller can be negative. Also the numbers don’t agree with the statement. The PID controller appears to have the highest R^2 (0.82) despite the statement say I is higher.

Coefficient of determination (R^2) is defined as

$$R^2 = 1 - \frac{SS_{res}}{SS_{tot}}$$

where,

$$SS_{res} = \sum_i (fit_i - data_i)^2$$

$$SS_{tot} = \sum_i (data_i - \overline{data})^2$$

Thus, R^2 can be negative if the model fit is bad, as in the case of the P controller, the error residuals (SS_{res}) are greater than the total sum of squares of the dataset.

The R^2 of PID is indeed higher, so we fixed the statement to

Lines 836-841: “Fits based on I, PI, PD, PID models were significantly different from the rest, out of which the I model was the most parsimonious based on the components in the system (Kruskal Wallis, Nemenyi test, $p < 0.01$; $n = 133$ trajectories; median values of R^2 – P: -0.61, I: 0.81, PI: 0.80, PD: 0.83, II: 0.63, PID: 0.81; median values of nAIC - P: 3.73, I: 1.40, PI: 1.43, PD: 1.29, II: 2.59, PID: 1.56)”

That said the goodness of fits of the models are not statistically different, and hence we chose I on the basis of parsimony.

I found the italics a bit over-used and distracting.

We have reduced the italics except in cases when we first introduce a new term.

Reviewer #2 (Remarks to the Author):

In this manuscript Natesan et al. explore the differential roles of two mechanosensory organs — Böhm’s bristles (BB) and Johnston’s organ (JO), in the control of antennal positioning. The work follows an earlier study from the same group (Krishnan et al. 2012) in which they established that BB, but not JO, are required for antennal positioning during flight, and suggest, due to short latency of activation, that BB sensory neurons may make direct synaptic contacts onto antennal motor neurons. The present study argues based on stabilization experiments that JO inputs determine the setpoint of a reflex mediated by BB. They fit the dynamics of antennal responses to mechanical perturbations with both a control theoretic model and a neural circuit model.

The questions raised by this study— how is input from different sensors combined to control a motor output— are of broad interest, and the system described here for studying them, in which two inputs can be isolated anatomically and the motor output studied directly, seems ideal for addressing these questions. The experiments that are performed appear to have been done carefully and rigorously. However, the broad conclusions of the paper as set out in the Introduction and Discussion are not fully supported by the experimental data presented. Only the JO are directly manipulated in this study, limiting the conclusions that can be drawn. In addition, while the modeling presents proof-of-concept that a simple neural circuit could give rise to the observations, it does not demonstrate that this is the circuit that actually underlies them, as no neural data are presented.

Many thanks for the positive comments about our experiments. We hope that the revised manuscript goes some way in highlighting that our focus is limited to just one other mechanosensor because we wish to set forth a methodology to incorporate more inputs in the future. In these regards, we agree that our paper is “proof-of-concept”, but hopefully that has not limited our ability to make the case.

Major concerns:

1) The study is motivated by describing how antennal positioning is modulated by multimodal cues, including wind, optic flow, and visual motion, as well as slower state-dependent cues such as flight versus walking. However, the study only examines the role of a single mechanosensory cue — wind speed — in flying moths. The study would be greatly enhanced by examining at least one other cue (flight state, optic flow, etc.) to see whether the same principles apply. For example, can all inputs be described as changing the antennal setpoint? Does the antennal positioning reflex operate in the same way when the moth is not flying?

Undoubtedly, the study would be greatly enhanced by adding more sensory cues. In principle, that is what our ultimate aim has been as outlined below. In practise however, incorporating each new cue is a major undertaking. Describing the Böhm’s bristle-motor neuronal circuit required basic studies of the circuit itself (Krishnan et al, JEB,

2012), a comparison of the Böhm's bristle-motor neuron circuit in honeybees, crickets and cockroaches (Sant and Sane, in press, 2019), and showing behavioural and neurophysiological evidence that antennal positioning was also influenced by optic flow in hawkmoths (Krishnan and Sane, JEB, 2014) and honeybees (Roy Khurana and Sane, 2016). This task itself took us nearly a decade. Our hypothesis from all these studies is 1) the antennal positioning reflex, consisting of Böhm's bristle-motor neuron reflex arc, is likely conserved in insects, and 2) that various sensory cues (antennal perturbation, optic flow, windspeed, olfaction etc.) alter the antennal setpoint *via* the motor neurons.

As a logical next step, we designed a minimal set of experiments that can be used to understand the interactions between the antennal positioning reflex, and the set-point modulation circuit. In this paper, we explore it for JO-mediated wind speed cues, for which we have specific experimental data. These data provide some key insights about how fast behaviors may be composed of a fast but rigid reflex component, in addition to a slow but variable modulatory component. Together, these two components ensured a fine-tuned yet rapid response.

Moreover, our attempt is to adopt the methodology or "language" of control theory to formally describe such neuroethological processes. In due course, as we increase our resolution of various other inputs into the antenna-motor circuit, we would extend this conceptual framework to incorporate those influences.

The antennal positioning reflex is only operational when the insect is in "flight" mode—either when warming up to get ready for flight, or else when flying. Our hunch is that the reflex operates the same way in both cases.

2) The study focusses on the interaction between BB and JO mediated signals, but only JO is manipulated in the experiments. The conclusions about the role of BB and the circuit underlying its action rest entirely on the previous paper (Krishnan et al. 2012). It would be helpful for the authors to clarify how the experiments described in that paper (Figure 2) differ from those shown in Figure 1 of the current manuscript. At least it seems like the authors should repeat the main experiments shown here (antennal response to different windspeeds and to perturbations at different wind speeds) in BB-stabilized moths to directly compare the roles of the two organs.

The experiments from Krishnan et al 2012 are fundamentally different than those conducted here. In those experiments, the focus was on BB and its role in antennal positioning, whereas here we focus on JO – which is the next logical step after BB. Behavioural experiments in Krishnan et al. 2012 quantified the effect of bristle ablation in antennal positioning. They showed that Böhm's bristles are necessary for proper positioning of the antenna; ablation of these bristles cause a loss in ability to position antenna. Additionally, they show that this positioning is likely mediated by a rapid, monosynaptic reflex loop. Here, we have addressed how such a rapid feedback loop interacts with modulatory inputs from the Johnston's organ. Because Böhm's bristle ablation causes a loss in antennal positioning, we could not perform experiments

involving antennal set-point modulation using these moths. Instead, we used electromagnets to modify the proprioceptive feedback from Böhm's bristles to quantify the interaction between the antennal positioning reflex and the set-point modulation circuit.

We have now modified the results section with the conceptual model to better clarify how the experiments from Krishnan et al. 2012 differ from and lead to the experiments performed in this paper. It now reads as follows.

Lines 146-184: "The results described above are consistent with previous data on airflow-dependent antennal positioning in honeybees [4], and hence indicate a mechanism that may be evolutionarily conserved. Although mechanosensory inputs from the Johnston's organ control antennal response to airflow, restricting these inputs by gluing the pedicel-flagellar joint does not affect the ability of the animal to position its antennae. Additionally, previous data on honeybees show that antennal response to other modalities, e.g. vision, is not affected in Johnston's organ restricted animals [4]. Therefore, initiation and maintenance of antennal positioning itself seems to be independent of the Johnston's organ mediated changes in antennal position.

In addition to sensory inputs from the Johnston's organ, the antennal motor system also receives proprioceptive inputs from the antennal hair plates (Böhm's bristles) [11,12]. Movement of the antenna stimulates neurons underlying these sensory hairs, which in turn activate the antennal motor neurons and the associated muscles with latencies on the order of 10 milliseconds [11]. Moths lose their ability to position their antennae when their Böhm's bristles are ablated, underscoring their importance in initiation and maintenance of antennal position [9–11]. This suggests that maintaining a stable position of the antenna requires mechanosensory inputs from the antennal hair plates. Restricting mechanosensory inputs from the Johnston's organ by gluing the pedicel-flagellar joint, on the other hand, does not affect maintenance of stable antennal position, but disrupts airflow-dependent antennal movements.

These and previous results enable us to construct a conceptual model of antennal positioning behaviour (Fig 2A). In this model, antennal position is encoded by the antennal hair plates which rapidly activate antennal muscles *via* a reflex arc [11]. This reflex operates as a negative feedback loop to ensure the initiation and maintenance of antennal position during flight. Johnston's organ mediated airflow-dependent changes are carried out by modulation of the set-point (equilibrium position) of this negative feedback loop. Ablation of hair plates, therefore, would break the feedback loop and cause a loss in the ability to position antennae. On the other hand, restriction of sensory inputs from Johnston's organ would still allow initiation and maintenance of antennal position, and only cause cessation of airflow-dependent movements.

More generally, we propose that antennal positioning behaviour comprises two sub-circuits (Fig 2A); one that maintains the antennae at a preferred position or set point using proprioceptive feedback from antennal hair plates (henceforth *antennal*

positioning reflex) and the other that modulates the set point using sensory inputs from multiple modalities, including airflow (henceforth *set point modulation circuit*). In the subsequent sections, we test this conceptual model using a combination of experiments and computational models. We changed the proprioceptive feedback from the hair-plates by perturbing antennal positioning using electromagnets (Fig 2B). We could not perform any hair plate ablation experiments because completely removes any proprioceptive inputs and does not change the feedback during experiments. Inputs to the set-point modulation circuit were experimentally altered by changing airflow and restricting the Johnston's organs (Fig. 1A). We hypothesize that airflow-dependent antennal positioning results from these two components working in concert, which allows the set point of the feedback loop to be tuneable.”

3) Two models are presented— a control theoretic model and a neural circuit model. In the case of the control theoretic model, different forms for the feedback term are explored, but different control architectures are not. That is, the model assumes that BB provides rapid feedback and JO alters the setpoint of the feedback. The neural circuit model contains several details about the structure of the neural circuit involved but is not supported by any data. It is nice to see that the model works but this does not demonstrate that this and not some other circuit architecture underlies the observed behavior. I think the modeling would have a greater impact if the authors could use this to show that their interpretation— that JO is determining the setpoint of a rapid BB-mediated reflex— is the only one consistent with the data.

In fact, our interpretation is supported by the neuroanatomical and neurophysiological data which were detailed in previous papers from our lab, which was perhaps not clear in our earlier text. We have rewritten sections of the manuscript to better clarify the details of the neural circuit model:

Lines 265-276: “The connectivity of the minimal neural circuit is based on electrophysiological and neuroanatomical data from previous studies which show that mechanosensory neurons of the antennal hair plates activate motor neurons of antennal muscles, likely *via* direct connections [11,12]. Therefore, in the minimal circuit, mechanosensory neurons were treated as simple on-off neurons that monosynaptically activate antennal motor neurons (Fig. 5A, see Methods). On the other hand, mechanosensory inputs from the Johnston's organ do not appear to form synapses with the motor neurons [12]. Thus, antennal motor neurons receive inputs either directly from primary sensory afferents or integrate inputs *via* interneurons that encode information from other modalities, such as vision, odour etc. For the sake of simplicity, we have assumed this to be direct in our minimal circuit model (Fig. 5A, see Methods). Finally, we modelled the motor neurons as simple integrate-and-fire neurons that pool incoming activity and control antennal muscles, and thus antennal position (Fig. 5A, see Methods).”

Strengthening the rationale for model neural circuits in the discussion.

Lines 361-367: “Based on the linear models and the existing anatomical and physiological data, we proposed a minimal neural circuit model that could maintain and modulate position based on context. Because a whole group of models were able to fit and predict the error correction dynamics of the antennal positioning reflex, several neural circuits are possible. The integral model was used based on the parsimony of components and formed the basic framework for higher order models. We used this control theoretic approach to generate mechanistic hypotheses of underlying neural circuits that perform this computation.”

Additionally, details of underlying assumptions in the minimal neural circuit have been added to the methods.

4) Several speculative statements in the Introduction and Discussion are presented without proper citations or data to back up these ideas. For example, lines 43-49 and 50-51 make multiple claims about the antennae and flight control without references.

We have tried to be as conservative as possible with our interpretations, but it is possible that the paper contains some aspects that are speculative, because it is partly theoretical. We have reviewed the introduction and discussion and added references wherever we could.

Minor concerns:

It is not clear from the Results and Figure legends that all data are from flying flies. This should be made clear.

We have now clarified in both the results and the figure legends that the data are collected from tethered moths that in ‘flight mode’, i.e. their antenna were positioned and they were flapping their wings.

The data shown in Figure 2C-F and Supp. Fig. 2 D-G are key to the paper. It would be helpful to show both of these in main figures so reader can directly compare them. Currently the supplementary figures lack figure legends, making the colors difficult to interpret. These same traces show intriguing dynamics that are not discussed in the text. For example the yellow trace in Fig. 2E shows some slow dynamics in the return to baseline. The authors should comment on this and note how these kinds of dynamics impact the interpretation of their model.

This was an excellent suggestion. As recommended, we have brought the JO-restricted electromagnetic perturbations to the main figures. This is now Fig. 3 of the paper. We have added the appropriate legends, and incorporated comments on dynamics. The raw data description of the new legends for both these figures read as follows:

Fig. 2C-F: “Response to perturbations in control moths. Representative raw data plots of antennal response to perturbations. (C) The right antenna (internal control) was unaffected by the perturbations of the left antenna, and its position depended only on the frontal airflows. (D) Azimuth-elevation plots show the clustering of right antennal position based on airflow. (E) When the electromagnet was on (grey), the left antenna was perturbed to a different angle, which was actively corrected on electromagnet release. The corrected angle depended on the frontal airflow. Sometimes the moths varied their corrected position during trials. An example of this is the response of the representative control moth at 1.5 m s^{-1} . Such changes may arise due to modulations in set-point owing to other modalities (Fig. 2A). (F) Five distinct antennal position clusters were observed of which four corresponded to the subjected airflows, and the fifth to the perturbed location.”

Fig. 3A-D: Response to perturbations in Johnston’s organ restricted moths. Representative raw data plots of antennal response to perturbations. (A) The right antenna (internal control) was unaffected by the perturbations of the left antenna, same as in control. (B) Azimuth-elevation plots show the clustering of right antennal position based on airflow. The clusters for 0 , 1.5 and 2.5 m s^{-1} were identical. At higher airflows (4 m s^{-1} , purple trace) the antenna was unable to maintain the same position and slightly drifts backward. (C) The perturbed antenna was actively corrected in Johnston’s organ restricted moths. They, however, were corrected to the same position regardless of airflow (except for 4 m s^{-1}). The sudden spikes in between were caused by the movement of the antenna by the ipsilateral front leg. (D) Three distinct antennal position clusters were observed – two based on airflow (0 , 1.5 and 2.5 m s^{-1} form one cluster and 4 m s^{-1} forms the other one.). The third was the perturbed location.”

line 69: please clarify and justify what is meant by “trade-off”. Also in Discussion line 373.

We have modified these sentences to read as follows:

Lines 66-71: “Maintenance of antennal position thus involves a component that ensures easy movement of the antenna, and another that restricts its mobility. To examine how these two counteracting components ensure positional control, we investigated the *antennal positioning behaviour* in the hawkmoths. To maintain a stable antennal position, diverse insects require mechanosensory feedback from antennal hair plates (also called Böhm’s bristles in moths) [11–14].”

In Discussion, the new text now reads.

Lines 413-419: “Antennal positioning, as examined in this study, is an example of state-dependent behaviour in flying insects. The position at which the antenna is held impacts the acquisition of both mechanosensory and olfactory cues. This sets up a potential trade-off in which increasing sensitivity of one cue causes a loss in sensitivity of the other. For instance, active forward movement of the antennae with increasing airflow may help restrict the flagellum to operate in the linear range of the pedicel-flagellar

joint, thus enabling reliable acquisition of airflow-related and other flagellar vibrations [30,31].”

lines 122 and 131: maybe these don't need to be separate section headings?

Agreed. We have combined the section headings as suggested. The new heading reads:

Lines 121-122: “*Flying moths use mechanosensory feedback from Johnston’s organs to position their antennae in response to frontal airflow*”

lines 152-153 “Ablation of the hair plates” This statement needs a reference.

and

lines 152-155 How does “stable antennal positioning” differ from “airflow-dependent antennal positioning.” Do the Böhm’s bristles play a role in one context, but not in another?

We have modified this sentence to increase clarity. It reads as follows:

Lines 158-163: “Moths lose their ability to position their antennae when their Böhm’s bristles are ablated, underscoring their importance in initiation and maintenance of antennal position [9–11]. This suggests that maintaining a stable position of the antenna requires mechanosensory inputs from the antennal hair plates. Restricting mechanosensory inputs from the Johnston’s organ by gluing the pedicel-flagellar joint, on the other hand, does not affect maintenance of stable antennal position, but disrupts airflow-dependent antennal movements.”

line 179/Figure 2: It might be easier for the reader if the data were presented first, followed by the conceptual model. In addition, the control theoretic model in Figure 3A could be arranged graphically in the same way as the conceptual model in Figure 2A.

We have modified the arrangement of Fig. 2A and Fig. 3A (now Fig. 4A) to make them graphically consistent.

line 190/Figure S2: The set point is not always the same in these data. Is this because the wind passively deflects the antenna to a different position? Please provide an explanation.

We added the below line to clarify increase in set-points in JO-restricted moths at high airflows.

Lines 207-211: “The antennal set-points for Johnston’s organ-restricted moths remained constant for low airflow values, and sometimes increased for high airflows (4 ms^{-1}), likely due to aerodynamic drag.”

line 240: linear integral system. Is it clear that this integration time reflect neural control rather than passive mechanical properties of the antenna? What happens in a dead fly?

We have added the below line to clarify that the correction is due to active neural control. Antennae in dead moths are stiff and do not exhibit the same properties as seen in live antennal positioning moths.

Lines 248-250: “Additionally, the consistent dynamics in error correction irrespective of the set-point suggest an active, rather than a passive, mechanism. “

line 246: the neural circuit model assumes that the MN itself determines the setpoint but no data are presented to support this hypothesis. Could it not be set somewhere more centrally?

Yes, it can be set centrally as well. We have modified this sentence to reflect this. It reads as follows:

Lines 269-276: “On the other hand, mechanosensory inputs from the Johnston’s organ do not arborize directly onto the motor neurons [12]. This transmission could be directly from primary sensory afferents or could occur via the higher centres where information from other modalities, like vision, odour etc, are integrated with this information. For the sake of simplicity, we have assumed this to be direct in our minimal circuit model (Fig. 5A, see Methods).”

line 287: does airflow-dependent modulation of antennal position occur in non-flying flies? If not, what does this tell us about the neural control of this reflex?

In resting moths, the antenna is tucked behind the wing and quite inactive. Such moths do not show any difference in antennal positioning regardless of stimulus. In moths that are flying or preparing to fly (a process in which they actively flap their wings or “shiver” to increase thoracic temperatures), antennal positioning is present and likely governed by the same mechanisms, as during flight. Thus, the underlying neural control in this latter scenario is identical for flying vs. non-flying moths.

line 297: can the setpoint of the antenna be altered by other means (visual, odor, flight state) if JO is ablated?

Yes, it can be altered by other modalities even in the JO-restricted moths or honeybees (see Khurana and Sane, 2016). We have modified the sentence to reflect this.

Line 318-322: “When these modulatory inputs are reduced (e.g. by restricting the Johnston’s organ), the moths retain their ability to maintain their antennal position at an arbitrary set point, but do not alter the set point at which the antenna are held (Fig. 3). For instance, Johnston’s organs-restricted honeybees retain the ability to modulate antennal position based on optic flow [4]”

line 314/figure. 3C: could these curves be fit with a single exponential decay? What do these curves look like if they are overlain and how do these compare to a single exponential decay? Is the control theoretical model necessary to account for these or could passive mechanical properties play a role?

Yes, they can – which is what the Integral model gives. However, they cannot be fit to a single exponential decay for every airflow case, and it is necessary to specify a set-point for this to work. We argue that this requires active control. To clarify the necessity of control theoretic model to account for these:

Lines 337-348: “We used standard linear models to fit the error correction dynamics of the antenna and found that the dynamics depended only on the error between current position and the set-point. The dependence on error in position instead of absolute position indicates that an underlying mechanism of active error correction rather than a passive mechanical rebound. The linear control theoretic models $L(s)$ captures the dependence of output position with respect to this error, with the integral model mathematically equivalent to a decaying error exponential. Other higher-order models (PI, PD, PID) can be used to capture more complex dependencies between the output and the error. We provided different amplitudes of step perturbations, but this disturbance was insufficient to distinguish the difference in performances of the higher-order models. Other stimuli (e.g. sum of sines, white noise or chirps etc.) may likely provide better resolution for which of the four models (I, PI, PD, PID) best approximates the error correction dynamics of the antenna under a variety of disturbances [32–34].”

353-354 I believe that the DUM innervates antennal muscles directly in some insects as well (Braunig et al. 1990, Allgauer & Honegger 1993).

Thanks for this point. We have altered the sentence and added the citations to incorporate this point.

Lines 394-397“ For example, in crickets inhibitory or excitatory Dorsal Unpaired Median (DUM) neurons symmetrically innervate the antennal muscles on both sides [6, 37–39].”

Reviewer #3 (Remarks to the Author):

This paper examines the feedback control system that underlies how moths regulate antennal angle with respect to airflow during flight. While the scope of the paper is narrowly focused, the studied reflex has important behavioral implications to flight stability, state-dependent sensory tuning, and multi-sensory integration. The main result of this work delineates the contributions of two separate antennal mechanosensory structures: the Johnston’s organ (JO) and Bohm’s bristles. The JO mediates antennal repositioning in response to changes in air flow; Bohm’s bristles provide the fast sensory cues that help regulate the antenna to the set-point angle. The authors present a thorough and convincing analysis, incorporating elegant tethered-flight perturbation experiments, a comparison of control theoretic models fit to data, and a proposed neuronal instantiation of the model. Such models may also inform understanding of other mechanosensory-motor feedback loops in insects, such as those that regulate wing or leg kinematics.

Additionally, the manuscript is thorough and well-written and the figures are clear and informative. I have no major concerns regarding this manuscript, so below I outline my moderate and minor comments.

Many thanks for the positive and supportive feedback. We are very happy that the paper was clear, as we hope the revised version is too.

Moderate comments:

Regarding airflow, antennal position, and drag force:

You report that for a given airflow, moths exhibited a distribution of set angles (Lines 125-126 and Figure 1 D, E, F). If we consider the antenna as a beam, its deflection would be proportional to drag force. But at different antennal positions, the wetted area will be proportional to $\sin(\theta)$ and the aerodynamic drag on the antenna would have a $(u \cdot \sin(\theta))^2$ term.

Yes, indeed this may result in some of the variability that we see in our data.

2. Each antennal position changes the sensitivity of the sensor, airflow/force or airflow/deflection depending on what quantity the JO is measuring. Might moths be regulating the sensitivity of their airflow sensor, with individuals having different preferred sensitivities? Rather than plotting change in antennal position vs airflow, it might be interesting/informative to plot sensitivity (raw or baseline subtracted) vs airflow.

In accordance with this suggestion, we have plotted sensitivity, and added it to Supplementary Figure 2. Moths are indeed more sensitive at lower airflows, which saturates as airflow increases.

3. At least, you should show a distribution of antennal baselines at 0 m/s. From Fig 1D, some moths exhibit nearly 90 deg motion towards the center at the highest air flows! Does this suggest that the inter-antennal angle would be 180 deg (at least)?

Typically, the initial inter-antennal angle is well below 180, with the median ~120 degrees. To address the referee's concern, we have added the initial inter-antennal angles of moths in all three treatments to Supplementary Figure 2 as well.

4. In the Methods section (Lines 515-517), you mention that the head is free to move. Head movements would also affect the relative angle of the antenna with respect to air flow. Could you provide a distribution or approximate range of head movement? Are head movements significant enough to influence the proposed model?

Head movements do indeed affect the relative angle of the antenna with respect to airflow, but these seem small enough to not alter the airflow-dependent antennal response much, as the medians are typically centred around zero with the extremes reaching till around 20 degrees. Additionally, the antennal angle calculations are not

affected as they are calculated based on a head-centric definition. We have provided an additional supplementary figure with the distribution of head movements and added the following description to the text.

Lines 583-591: “Individual antennal angle was defined as the angle between an antenna vector and the head vector. Because the head is free to rotate in our experiments, a head-centric definition allowed us to compute the antennal angle independent of head rotation (Fig S5A-D). Due to electromagnetic perturbations, we observed small changes in head orientation throughout our experiments (Fig. 5E-G). Despite this, the right (control) antennal angle typically remained constant throughout the trial, suggesting that the head-centric method used to compute antennal angles eliminated the rotations of the head (Fig. S5A-D). Additionally, both active head rotations and the ones elicited by electromagnetic perturbation seem to be too small to affect airflow-dependent antennal positioning (Fig. S5A-D)”

Regarding parametrized control theoretic models:

Please report the best-fit parameters for the different model structures. In Figures 3B and 4D, the PI and I models coincidentally yield fits with identical goodness (down to the 0.01%). Is the P-value negligible? That is, did the proportional-integral model in effect reduce to the integral model? If so, that is an even stronger case for the I-model than just parsimony.

We have added Table-1 which contains the mean fit parameters for all the control theoretic models.

Minor edits

Line 45: “For optimal acquisition of sensory information and robust flight control...”

Firstly, we might suspect that’s why insects move their antennae, but I don’t know of evidence of this in the literature (if there is, it should be cited). Secondly, in the context of feedback control (mentioned in the second clause), “optimal” and “robust” have specific meaning. This paper does not evaluate optimality or robustness. I would weaken this clause, perhaps:

Insects actively move and position their antennae and eyes via fine feedback control of antennal muscles and head/eye movement, putatively to tune sensory sensitivities and stabilize flight.

We have modified the sentence as suggested. It now reads:

Lines 44-48: “On stroke-to-stroke timescales, flying insects rely heavily on rapid mechanosensory feedback from their antennae and visual feedback from their compound eyes [1–3]. They actively move and position their antennae and eyes *via* fine feedback control of antennal muscles and head / eye movements which aid in tuning the acquisition of sensory information and stabilizing flight [4,5].”

Line 53, 63 382: again, “optimal” and “optimizes” is over-reaching.

Agreed – we have no evidence of optimization. We have removed reference to optimal or optimizing wherever possible. We have retained it in a couple of places where we felt it was appropriate – but only in presenting a hypothetical scenario.

Line 161 (typo): “comprises of two sub-circuits” delete “of”

We have corrected this in the text.

Line 181-183: “Thus, ...” This sentence seems like an overly wordy way of saying that the antennal angle changed due to airflow. What is the use of explicitly enumerating the motor neurons, motor commands, and antennal muscles that effect the change?

We have changed this sentence. It now reads as follows

Lines 198-199: “Thus, the antennal set-point is altered by the frontal airflow, and actively maintained during flight.”

Line 200: “control theory model” should be “control theoretic model” or just “control model”

Optimal and robust

Neuronal network or neural circuit, so as to disambiguate from artificial neural network

As suggested, we have replaced “control theory model” with “control theoretic model” throughout the manuscript.

Line 292, 323, 325, 329, and elsewhere: You describe your model as a neural network which sets up expectations of an artificial neural network. To avoid confusion with the oh-so-popular machine learning tool, I’d substitute either “neuronal network” or “neural circuit”. Also, your model comprises very few cells; I think “circuit” is more accurately descriptive than “network”.

We agree that our model is better described as a “neural circuit” model. As suggested, we have replaced “neural network” model with “neural circuit” model throughout the manuscript.

Line 298: In referring to JO-inhibited preparations, “but cannot alter the set point.” Cannot or do not? JO-inhibited moths do not alter the set point in response to air flow, but they might still alter the set point if presented with an optic flow stimulus, no? So they physically could, but they do not because of the sensory inhibition.

We have fixed this as suggested. The modified sentence now reads as follows:

Lines 319-322: “When these modulatory inputs are reduced (e.g. by restricting the Johnston’s organ), the moths retain their ability to maintain their antennal position at an arbitrary set point, but do not alter the set point at which the antenna are held (Fig 2H, S2D-F).”

Line 383: delete “such”.

We have corrected this in the text. The new line is as follows.

Lines 426-427: “The linear integration model proposed here suggests a specific mechanism by which multiple cues may be integrated by the antennal motor neurons.”

Reviewers' Comments:

Reviewer #1:

Remarks to the Author:

The authors have addressed most of my comments with their detailed response. I especially appreciate their efforts to address my concerns on the timecourses, fitting, and posing of the question.

Overall I think the paper really motivates more rigorous system identification and model to better distinguish between the control models. I remain unsure if the Integral model fits best because it is truly sufficient to capture behavior under changing conditions or because the single step impulse could not distinguish strongly between the models. However, I do think that the data collected support the conclusions that the authors draw and the hypotheses poised with the modeling give a rich framework for future testing. The interaction of BB and JO and how that leads to a context dependent controller is a very nice articulation of the problem and it goes far in considering how inner and outer loop control needs to be considered carefully in control of biological movement. Further testing of this hypothesis would require new experiments, which would make for good follow-up literature.

One aspect that is still not coming across is how the authors address the interacting time scales. The model treats the set-point adjustment as a non-dynamical process – that is the authors shift the set point in experiment and model and then allow test the inner loop dynamics. There is nothing wrong with doing this, it is a good way to isolate the inner loop, but it does not fully test if timescales are separable -- there is not a dynamic systems description of the outer loop. I agree that the authors can discuss the antennal positioning response being nested inside the context-dependent response, but these timescales may not be fully separable as seems to be suggested and I think they should acknowledge this more clearly in the discussion if only to prompt the appropriate follow-up experiments. Untangling this could be done by fitting a controller to the outer loop in addition to the inner loop control and then perturbing both antenna position and windspeed, which would also provide a nice test of the circuit-based model. I'm not suggesting that this be done in this paper, just that something along those lines is needed to test the hypothesis that the time scales are fully separable. For now, I encourage the authors to maintain the point the one response is faster and one slower, but point out that these dynamics may interact during flight especially in variable airspeed conditions. Indeed this suggests that if airspeed is changing quickly, which one might expect maneuvers, then the antenna positioning may not be able to maintain an effective set-point.

When I suggested fitting a delay, I meant a free delay parameter. I think we were talking past each other bit. The authors made it clear why it might be hard to measure a delay in their system, but if they are fitting a transfer function with variable poles and zeros, I think they can still add an $\exp(-nT)$ term with n as a free delay parameter to their fits (again following Madhav, et al 2013). However, after the revision I think it is pretty clear that these models will also fit the data basically equivalently, because the simple I controller does about as well as one might expect at least for the step responses.

Line 126 The word "stabilized" is confusing because the antenna position is stabilized at each set point. I think you mean saturated or the like.

Reviewer #2:

Remarks to the Author:

The authors incorporated some of the constructive suggestions from the reviewers into the

manuscript, including some additional analyses, references, method details, and by moving a key supplemental figure into the main figures. The manuscript has been improved by these changes, however I have several remaining concerns that I outline below.

Major concerns:

1) The authors did not attempt any of the more substantial modifications recommended by the reviewers to strengthen the paper. Specifically, they do not attempt to model any other circuit architectures (for instance to examine the alternate hypothesis in which JO directly synapses onto antennal motor neurons) or add any additional experimental data requested. Further, the authors make a point of emphasizing that the paper is very theoretical, implying that no further experiments or citations are needed to support numerous speculations in the text (author response to Reviewer #2 major concern 4). I feel that these speculative statements and framing detract from the experiments and analyses they did perform.

2) The authors do not address the slow return to baseline noted by two reviewers (Figure 2E) anywhere in the text or formally with the model. Given that one of the major points of the study relates to the temporal dynamics of the antennal sensory motor circuit, this should be addressed.

3) In the Discussion, the authors are still lacking references to studies in which mechanosensory and olfactory cue acquisition is impacted by the set position of the antennae (Lines 413-419). Further, it is unclear what the "trade-off" is here – are the authors suggesting that the antennal flight position diminishes the animal's olfactory capabilities? If this is supported by any published evidence, then this may be a valid point, but the relevant study should be referenced here, and the details of this trade-off be laid out more clearly.

Minor concerns:

1) The authors never mention that the antennal positioning reflex is only observed in flying moths. This is an important point and should be mentioned in the introduction.

2) The authors do not show an average antennal response in Figure 2 (or 3), only a raw trace. In their comments to reviewer #1, they note that there is considerable trial-to-trial variability in the antennal response. This is not, however, clear from the figures (2 and 3), and is concerning. Perhaps the authors could include part of the raw trace already shown in Figures 2 and 3, and an additional trace that depicts the average plus other individual traces, to depict the full range of responses?

3) In the introduction, the authors describe the antennae as moving in a "context-specific manner" in which the contexts are different behaviors (walking, flying, foraging, escaping, etc.; line 60). This study focuses on a single behavioral context: flight. Later in the manuscript, the authors discuss context dependent positioning of the antennae (for example line 292), where the context is airflow speed. It would be helpful to perhaps use the term "context" for only one of these categories – behavior (walking vs. flying etc.) or specific sensory experience (airflow) to avoid confusion.

4) Line 66 (added in the revision): Can you be more specific here? What do you mean by "easy" movement?

5) Line 162 – reference needed.

6) Line 191-192: The authors claim that the right antenna was unaffected by magnetic perturbations of the contralateral antenna. However, I do notice a subtle decrease in angle during the magnetic

perturbation in the raw trace shown, and I wonder if this change occurred in the other experimental animals. I do not think this observation casts significant doubt on the overall findings of this experiment, but it does suggest that an average antennal trace (across multiple animals; see Major concern #3, above) would be useful for interpreting this data.

7) Line 360 – references should be included for the existing anatomical and physiological data.

8) Line 893 Subfigure should read (D). The additional figures in Figure S2 are useful, however it looks like a statistical comparison was left out. The authors discuss differences in sensitivity at low and higher airflow rates (lines 133-135), however it is unclear whether these are statistically significant differences.

Reviewer #3:

Remarks to the Author:

Largely, I was happy with the paper as it was originally submitted. But I realize I may not have been clear enough in my first few comments, because I feel the authors missed the point in how these issues/recommendations were addressed.

Regarding the outer control loop (mediated by the JO), the authors report antennal set point angles described in baseline-subtracted, moth-head-centric coordinates. My first comment prodded the relationship between set-point angle and the drag force on the antenna (which is a non-linear function of air flow and antennal angle in air-flow-centric coordinates). I should have been more explicit that the authors should consider if there is a functional relationship between the airflow, antennal angle and drag force, ie plot the drag force vs airflow. My subsequent comments about sensitivity analysis (how does drag force change with change with air speed about different antennal set points) and head angle all relate to this representation.

I was trying to motivate a line of inquiry: is there some representation/transformation of the input and output for which the changes in antennal angle preserve some physical quantity. That is, the outer loop (JO) is not just a look up table of airflows and antennal angles, but itself a control system that is regulating some measured signal (drag force) or system property (sensitivity to changes in airflow). For example, given the changes in antennal set point angle, if the mapping from airflow to drag force is flat, it would suggest that the JO loop is a control system that modulates antennal position to regulate drag force. If the plot of airflow vs sensitivity ($d \text{ Force} / d u$ evaluated at each angular set point) is flat, then you might posit that the outer loop modulates the sensitivity for the inner loop.

The data are all there; the suggested analyses just require a transformation of the data. As it is, the authors provide a description of the outer loop, the mapping of airflow to antennal set point as a look-up table (and again, I felt that this description was quite nice as it was). Representing this mapping in terms of force and/or sensitivity might reveal a more concrete functional role of the JO circuit, elevating the story further.

Additional comments regarding the edits:

There are still a few references to the “neural network” model, both in the Table and SI. These too should be changed to “neural circuit.”

In response to Reviewer 1, the authors added the sentence:

“So, any complex behaviour may be thought to contain a component that is fast and invariable, and

another that is slow but variable.”

This is an overstatement. There has not been previous discussions of complex behaviors in this paper (what constitutes a complex behavior) and no evidence that these nested slow and fast loops are universal to complex behaviors. I don't disagree with the sentiment, but perhaps soften the wording. I suggest something like “So, many seemingly simple reflexive responses may comprise circuits at different time scales: a component...”

Response to Reviewers' comments for NCOMMS-18-38698A

We thank all three reviewers for providing thoughtful feedback and for their continuing engagement with our paper. In this round of revision, they highlighted two main issues – interacting timescales and nested feedback loops - that required more clarification. These comments have especially helped put focus on the key “big picture” point - the hierarchical organization of behaviours within nested feedback loops –which, in retrospect, was under-emphasized in our manuscript, and especially in our abstract. On this issue, perhaps we erred on the side of being too conservative. We thank the reviewers for emphasising this aspect of the manuscript. Several other changes were also suggested which we have taken on board, and the manuscript underwent some language editing to fit it to word limit. Together, we hope these will make this manuscript acceptable for publication.

Responses to Reviewer #1:

The authors have addressed most of my comments with their detailed response. I especially appreciate their efforts to address my concerns on the timecourses, fitting, and posing of the question.

Overall I think the paper really motivates more rigorous system identification and model to better distinguish between the control models. I remain unsure if the Integral model fits best because it is truly sufficient to capture behavior under changing conditions or because the single step impulse could not distinguish strongly between the models. However, I do think that the data collected support the conclusions that the authors draw and the hypotheses poised with the modeling give a rich framework for future testing. The interaction of BB and JO and how that leads to a context dependent controller is a very nice articulation of the problem and it goes far in considering how inner and outer loop control needs to be considered carefully in control of biological movement. Further testing of this hypothesis would require new experiments, which would make for good follow-up literature.

This paragraph accurately highlights the main motivations of the paper, as well as the areas in which more efforts are required in the future. In this manuscript, we had hoped to present the antennal positioning response in insects as a case in point for how behavioural circuits are hierarchically organized, and we hope that this perspective guides future studies in other systems as well.

One aspect that is still not coming across is how the authors address the interacting time scales. The model treats the set-point adjustment as a non-dynamical process – that is the authors shift the set point in experiment and model and then allow test the inner loop dynamics. There is nothing wrong with doing this, it is a good way to isolate the inner loop, but it does not fully test if timescales are separable -- there is not a dynamic systems description of the outer loop. I agree that the authors can discuss the antennal positioning response being nested inside the context-dependent response, but these timescales may not be

fully separable as seems to be suggested and I think they should acknowledge this more clearly in the discussion if only to prompt the appropriate follow-up experiments. Untangling this could be done by fitting a controller to the outer loop in addition to the inner loop control and then perturbing both antenna position and windspeed, which would also provide a nice test of the circuit-based model. I'm not suggesting that this be done in this paper, just that something along those lines is needed to test the hypothesis that the time scales are fully separable. For now, I encourage the authors to maintain the point the one response is faster and one slower, but point out that these dynamics may interact during flight especially in variable airspeed conditions. Indeed this suggests that if airspeed is changing quickly, which one might expect maneuvers, then the antenna positioning may not be able to maintain an effective set-point.

We fully agree on the issue of interacting timescales raised by both reviewers (see also our response to Reviewer #2). The referee's point that we should highlight the faster vs. slower responses while allowing for interacting timescales during sudden perturbations, is well-taken. As suggested, we have added the following paragraph address this issue:

Line 349-359: "In our control model, set-point is assumed to be fixed for each value of airflow. This allowed us to isolate and characterize the inner loop dynamics, thereby identifying its stereotypic error correction (Fig. 4A). Because goodness of fits for these models were high (Fig. 4D-E), we inferred that the timescales of the inner loop were faster than those of the set-point modulation circuit. We concluded that the Böhm's bristle-mediated reflex loop rapidly maintains antennal position, whereas feedback from JO slowly modulates set-point based on airflow. However, airflow sensing by JO and resulting set-point modulation has its own temporal dynamics, which may interact with that of the inner feedback loop during flight. This may be especially true in variable airspeed conditions, e.g. rapid flight manoeuvres or a sudden wind gust. This motivates the need for experiments and modelling that are specifically targeted towards understanding how insect nervous systems disambiguate the interacting timescales of these two circuits."

When I suggested fitting a delay, I meant a free delay parameter. I think we were talking past each other bit. The authors made it clear why it might be hard to measure a delay in their system, but if they are fitting a transfer function with variable poles and zeros, I think they can still add an $\exp(-nsT)$ term with n as a free delay parameter to their fits (again following Madhav, et al 2013). However, after the revision I think it is pretty clear that these models will also fit the data basically equivalently, because the simple I controller does about as well as one might expect at least for the step responses.

We discussed this possibility a fair bit amongst us. We agreed that adding a free delay parameter would make the model more general but worried that it would also make it less parsimonious. Because much of our emphasis is on parsimony (all models being more or less equal in their ability to fit the data), we felt that a simple I-system seems at present to be sufficient to model the data that we have obtained.

Line 126 The word “stabilized” is confusing because the antenna position is stabilized at each set point. I think you mean saturated or the like.

Yes, we did indeed mean “saturated” and not “stabilized”. We have now fixed this.

Responses to Reviewer #2:

The authors incorporated some of the constructive suggestions from the reviewers into the manuscript, including some additional analyses, references, method details, and by moving a key supplemental figure into the main figures. The manuscript has been improved by these changes, however I have several remaining concerns that I outline below.

Major concerns:

1) The authors did not attempt any of the more substantial modifications recommended by the reviewers to strengthen the paper. Specifically, they do not attempt to model any other circuit architectures (for instance to examine the alternate hypothesis in which JO directly synapses onto antennal motor neurons) or add any additional experimental data requested. Further, the authors make a point of emphasizing that the paper is very theoretical, implying that no further experiments or citations are needed to support numerous speculations in the text (author response to Reviewer #2 major concern 4). I feel that these speculative statements and framing detract from the experiments and analyses they did perform.

In retrospect, our response to Reviewer #2’s major concern 4 may have come across as somewhat dismissive, which was not our intention. Our attempt has been to stay as close to the data as possible. The circuit architectures that we modelled were strongly rooted in neuroanatomical and physiological studies. For instance, our neuroanatomical data shows that axonal arbours of the Böhm’s bristles co-localize extensively with dendritic branches of the antennal motor neurons (as described in Krishnan et al, JEB, 2012; Sant and Sane, J Comp Neurol., 2018) and our neurophysiological data show very low latency responses of the antennal muscles in response to stimulation of Böhm’s bristles (Krishnan et al, JEB, 2012). These provided the structural basis of the inner loop constituting a sensorimotor reflex which operates in *very short* timescales.

In contrast, the hypothesis of JO directly synapsing onto the motor neurons is not supported by our neuroanatomical data, because the Johnston’s organ arbours do not co-localize with the dendritic arms of the motor neuron, although they too terminate in the AMMC (Sant and Sane, 2018). Indeed, there is a spatial separation between the two, which makes it highly unlikely that JO arbours directly synapse on the antennal motor neurons (e.g. see Fig 8, Sant and Sane, J Comp. Neurol. 2018). These data, in conjunction with the behavioural data set strongly suggest the possibility of JO action through pathways that act *via* interneurons, and hence operate over *longer timescales*.

We therefore hypothesize that the Bohm’s bristle mediated response loop and its modulation by JO-mediated feedback operate over different timescales. However, as

Reviewer 1 also observed, we acknowledge that there may be some overlap in the timescales of these two, especially during fast manoeuvres.

Reviewer 2's point that we should add references to back our statements is also well-taken. We have reviewed the manuscript and added references wherever relevant (examples below):

Line 50-51: "For instance, during flight, insects integrate rapid mechanosensory feedback from their antennae with slower visual feedback from their compound eyes [1-3]"

Line 51-53: "They actively position their antennae and eyes *via* fine feedback-control of their antennal muscles and head/eye movements, thereby tuning sensory acquisition for flight stabilization [4,5]"

The current model is based on quantification of timescales using the behavioural data. We agree with the reviewer that new (more physiological) experiments are required to quantify the precise timescales in which JO influences activity in the antennal muscles. This would allow us to add specific values in the next version of these models, but they are unlikely to alter the conclusions drawn here.

2) The authors do not address the slow return to baseline noted by two reviewers (Figure 2E) anywhere in the text or formally with the model. Given that one of the major points of the study relates to the temporal dynamics of the antennal sensory motor circuit, this should be addressed.

As Reviewer 1 also points out, it is likely that the timescales of the two loops may have some overlap – and we acknowledge that our model does not account for it.

We have now modified the section to clarify this.

Line 378-386: "The model neural circuit captures the flow of information from Böhm's bristles and JO to the antennal muscles. Böhm's bristles control motor neuronal activity *via* negative feedback (inhibition/antagonistic excitation), whereas JO modulates its set-point. The specific set-point modulation can happen in multiple ways, including activation of antennal motor neurons by JO either *via* interneurons or by direct synapses onto them, although the latter possibility is not supported by neuroanatomical data which show very little co-localization between axonal arbours of JO with dendritic fields of Böhm's bristles [15]. This modulation translates to asymmetric activation of antennal motor neurons by JO, which is a key feature of information flow from JO in the model neural circuit (Fig 5A)."

As suggested, we have now addressed the slow return to baseline in Fig. 2E in the discussion.

Line 401-407: "Altering the error correction dynamics requires changing the overall excitability of both muscles, which in turn requires symmetric excitation/inhibition of all motor neurons/muscles. For example, in crickets inhibitory or excitatory Dorsal

Unpaired Median (DUM) neurons symmetrically innervate antennal muscles on both sides [4,40–42]. Activity in these neurons may change the dynamics without altering set-point. Such changes might explain the slow return to baseline seen in a few trials (Fig 2E, Fig S6).”

3) In the Discussion, the authors are still lacking references to studies in which mechanosensory and olfactory cue acquisition is impacted by the set position of the antennae (Lines 413-419). Further, it is unclear what the “trade-off” is here – are the authors suggesting that the antennal flight position diminishes the animal’s olfactory capabilities? If this is supported by any published evidence, then this may be a valid point, but the relevant study should be referenced here, and the details of this trade-off be laid out more clearly.

We agree and have added references to studies. It now reads as follows:

Line 423-435: “Antennal positioning, as examined in this study, is an example of state-dependent behaviour in flying insects. Antennal position impacts acquisition of both mechanosensory and olfactory cues. For instance, active forward movement of antennae with increasing airflow may help restrict the flagellum to operate in the linear range of pedicel-flagellar joint, thus enabling reliable acquisition of airflow-related or other flagellar vibrations [30,31]. However, this may diminish the ability of antenna to sample the odour space around the insect as it decreases spatial sampling [49,50]. On the other hand, if antennae are held at a large angle to increase odour capture, aerodynamic drag increases thereby affecting its mechanosensory function [30,31]. This sets up a potential trade-off in which increasing sensitivity of one (say, olfactory) cue compromises sensitivity of the other (mechanosensory). In addition to mechanosensory and olfactory cues, antennal position is also influenced by visual feedback [9,19,20], which may impact both olfactory and mechanosensory feedback. Such trade-offs for acquisition of sensory stimuli have received very little attention.”

Minor concerns:

1) The authors never mention that the antennal positioning reflex is only observed in flying moths. This is an important point and should be mentioned in the introduction.

Thank you for noting this point. We have now added this to the last paragraph of the introduction.

Line 94-98: “Here, we address this question by investigating the neural principles underlying *airflow-dependent antennal positioning* in the oleander hawkmoth, *Daphnis nerii*. Because both stable positioning of antenna and airflow-dependent changes occur in preparation for, or during flight, our experiments were performed on tethered flying hawkmoths.”

2) The authors do not show an average antennal response in Figure 2 (or 3), only a raw trace. In their comments to reviewer #1, they note that there is considerable trial-to-trial variability

in the antennal response. This is not, however, clear from the figures (2 and 3), and is concerning. Perhaps the authors could include part of the raw trace already shown in Figures 2 and 3, and an additional trace that depicts the average plus other individual traces, to depict the full range of responses?

Following the reviewer's suggestion, we have now added a supplementary figure (Fig. S6) to include the raw traces for all individuals, along with an average trace.

The trial-to-trial variability mentioned in our response to Reviewer #1's query relates to the variability in the antenna release by the electromagnet – i.e. the delay in switching off the electromagnet and antennal movement due to the inherent delay associated with decay of the magnetic field produced by the electromagnet (~75 ms), combined with the variability in antennal morphology (antenna size, antennal muscle power, etc). Once free to move, the antennal response is very stereotypic (Fig. S6), as captured by the I-model.

3) In the introduction, the authors describe the antennae as moving in a “context-specific manner” in which the contexts are different behaviors (walking, flying, foraging, escaping, etc.; line 60). This study focuses on a single behavioral context: flight. Later in the manuscript, the authors discuss context dependent positioning of the antennae (for example line 292), where the context is airflow speed. It would be helpful to perhaps use the term “context” for only one of these categories – behavior (walking vs. flying etc.) or specific sensory experience (airflow) to avoid confusion.

Thanks for pointing this out. We now only use “context” to specify different behaviours. We have changed this throughout the text, as suggested, to avoid confusion.

Line 65-66: “Third, antennal movements are context-specific, and depend on whether the insect is walking, flying, foraging, escaping etc. [4].”

Line 284-286: “Thus, the realized neural circuit maps onto the control theoretic model, providing a mechanistic basis for airflow-dependent antennal positioning.”

Line 361-363: “Based on the linear models and existing anatomical and physiological data [14,15], we proposed a minimal neural circuit model that could maintain and modulate position based on sensory stimulus.”

4) Line 66 (added in the revision): Can you be more specific here? What do you mean by “easy” movement?

We have made the line more specific as suggested.

Line 71-73: “Maintenance of antennal position thus requires a behavioural module that ensures smooth, unrestricted movement of the antenna, while another module restricts its mobility.”

5) Line 162 – reference needed.

We have added figure references as suggested and modified the sentence to make it read better.

Line 162-164: “On the other hand, restricting mechanosensory inputs from the JO by gluing the pedicel-flagellar joint does not affect the moths’ ability to position their antenna, but disrupts airflow-dependent antennal movements (Fig 1F, Fig S2C, Supplementary Video 3)”

6) Line 191-192: The authors claim that the right antenna was unaffected by magnetic perturbations of the contralateral antenna. However, I do notice a subtle decrease in angle during the magnetic perturbation in the raw trace shown, and I wonder if this change occurred in the other experimental animals. I do not think this observation casts significant doubt on the overall findings of this experiment, but it does suggest that an average antennal trace (across multiple animals; see Major concern #3, above) would be useful for interpreting this data.

As suggested, we have now added the raw traces of left and right antennal angles for all the animals in Fig. S6. We find that, on average, the right antenna is unaffected by magnetic perturbations of the contralateral antenna, both in control and JO restricted moths. We have appended the text to incorporate the new supplementary figure and specify that we generally see no contralateral perturbation.

Line 191-192: “Average position of right (unperturbed) antenna remained unaffected by magnetic perturbations delivered to the contralateral antenna (Fig 2C-D, Fig. S6).”

Line 609-612: “Despite this, the average position of the right (control) antennal angle typically remained constant throughout the trial, suggesting that the head-centric method used to compute antennal angles eliminated rotations of the head (Fig. S5A-D, Fig. S6).”

7) Line 360 – references should be included for the existing anatomical and physiological data.

The references have been added.

Line 361-363: “Based on the linear models and existing anatomical and physiological data [14,15], we proposed a minimal neural circuit model that could maintain and modulate position based on sensory stimulus.”

8) Line 893 Subfigure should read (D). The additional figures in Figure S2 are useful, however it looks like a statistical comparison was left out. The authors discuss differences in sensitivity at low and higher airflow rates (lines 133-135), however it is unclear whether these are statistically significant differences.

We have added statistical comparisons for figure S2 both in the text and in the figure legends. Thanks for pointing this out.

Line 137-140: “Their inter-antennal angle remained unchanged at low airflow values (Fig. S2D, Kruskal Wallis test, $p=0.78$), and even increased slightly at higher airflow values, likely due to backward aerodynamic torques (Fig. 1F).”

Fig S2 914-917: “Sensitivity changes to airflow of control and sham-treated moths were correlated ($\rho=0.70$) whereas sensitivity of JO restricted moths were not correlated with the other treatments or with changes in airflow (with control – $\rho = -0.21$; with sham – $\rho = -0.09$; with airflow – $\rho = 0.04$).”

Fig S2 926-927: “(D) *Interantennal angle for no airflow (0 ms^{-1}) of all three treatments. The three treatments were not statistically different from each other (Kruskal Wallis test, $p=0.78$).”*

Responses to Reviewer #3:

Largely, I was happy with the paper as it was originally submitted. But I realize I may not have been clear enough in my first few comments, because I feel the authors missed the point in how these issues/recommendations were addressed.

Regarding the outer control loop (mediated by the JO), the authors report antennal set point angles described in baseline-subtracted, moth-head-centric coordinates. My first comment prodded the relationship between set-point angle and the drag force on the antenna (which is a non-linear function of air flow and antennal angle in air-flow-centric coordinates). I should have been more explicit that the authors should consider if there is a functional relationship between the airflow, antennal angle and drag force, ie plot the drag force vs airflow. My subsequent comments about sensitivity analysis (how does drag force change with change with air speed about different antennal set points) and head angle all relate to this representation.

I was trying to motivate a line of inquiry: is there some representation/transformation of the input and output for which the changes in antennal angle preserve some physical quantity. That is, the outer loop (JO) is not just a look up table of airflows and antennal angles, but itself a control system that is regulating some measured signal (drag force) or system property (sensitivity to changes in airflow). For example, given the changes in antennal set point angle, if the mapping from airflow to drag force is flat, it would suggest that the JO loop is a control system that modulates antennal position to regulate drag force. If the plot of airflow vs sensitivity ($d\text{ Force}/d\text{ u}$ evaluated at each angular set point) is flat, then you might posit that the outer loop modulates the sensitivity for the inner loop.

The data are all there; the suggested analyses just require a transformation of the data. As it is, the authors provide a description of the outer loop, the mapping of airflow to antennal set

point as a look-up table (and again, I felt that this description was quite nice as it was). Representing this mapping in terms of force and/or sensitivity might reveal a more concrete functional role of the JO circuit, elevating the story further.

The question posed here (“Why do insects move their antennae in response to airflow?”) is very important, but has proved quite elusive. The reviewer proposes a very specific hypothesis: do moths maintain the drag (or, rather, mechanical torque) on the antennae at a constant value? If yes, then we should calculate constant torques when the antennal angle changes in response to changing airflow. Exactly this question was addressed by Gewecke and Heinzel in their 1980 paper (this references is now added). Using their method, we calculated the torques on antennae under diverse airflow and antennal angles that we observed. In a nutshell: our results are the same as those of Gewecke and Heinzel: torque is somewhat mitigated, but not maintained at a constant value due the antennal repositioning.

The answer to this question is likely more nuanced because antennal response is also a function of other inputs such as optic flow (Khurana and Sane, eLife, 2016; Krishnan and Sane, JEB, 2014). So there are conditions under which for the same airflow, antennal angles may be modulated using changes in optic flow. This makes the question of why insects move their antennae in response to these stimuli much harder to address.

Line 298-303: “Antennal positioning at flight onset and its airflow-dependent modulation have been observed in diverse insects (honeybees, locusts, flies: [17,18,20]). Airflow-dependent modulation is mediated by the JO, which likely senses antennal deflections due to aerodynamic torques [4,30,31]. How this behaviour aids in sensory acquisition remains as yet unclear; airflow-dependent modulation does not seem to maintain aerodynamic torques (Fig. S2A-C, also [30,31]), and depends on multiple sensory inputs including airflow and optic flow [9].”

Also-

Line 548-557: “The sensitivity of antennal movements to airflow was computed as change in interantennal angle per 0.5 ms^{-1} step change in airflow (Fig. S2A-C). The torque on the antennal base due to aerodynamic drag was non-linear and dependent on the antennal angle and speed of frontal airflow [30]. It was computed using an equation (derived from [30]):

$$T_{\gamma} \sim \sin \gamma \cdot v_a^{1.44}$$

in which γ is the antennal angle and v_a is the speed of airflow. T_{γ} was calculated for all three treatments (using data in Fig. 1D-F). If feedback from JO is regulates torques by changing antennal position, T_{γ} should remain constant for increasing airflows. Airflow-dependent changes in antennal position mitigated T_{γ} (Fig. S2A-C; compare control, sham with JO restricted), but did not maintain it at a constant value.”

Fig. S2:

Fig S2 918-924: “(A-C) (iii) *Torque on the antennal base due to aerodynamic drag in control (A), sham-treated moths (B) and JO restricted moths (C).* Torque on the antennal base (T_γ) had non-linear dependence on antennal angle and airflow speed (equation derived from [30], see methods). Modulation of antennal angle in control and sham-treatment moths reduced the slope of T_γ for increases in airflow, in comparison to JO restricted moths. However, T_γ increased with airflow for all three treatments (average slope for control: 45° , sham: 47° , JO-restricted: 62° . The slopes were from linear fits with $\text{adj}R^2 > 0.95$.”

Additional comments regarding the edits:

There are still a few references to the “neural network” model, both in the Table and SI. These too should be changed to “neural circuit.”

We have changed this as suggested.

In response to Reviewer 1, the authors added the sentence:

“So, any complex behaviour may be thought to contain a component that is fast and invariable, and another that is slow but variable.”

This is an overstatement. There have not been previous discussions of complex behaviors in this paper (what constitutes a complex behavior) and no evidence that these nested slow and fast loops are universal to complex behaviors. I don’t disagree with the sentiment, but perhaps soften the wording. I suggest something like “So, many seemingly simple reflexive responses may comprise circuits at different time scales: a component...”

We agree. We have softened the wording as recommended by the reviewer. Reviewer #1’s comments urged us to see these behaviours as nested “loops” – this idea was implicit in our first version, and we felt (after reviewer #1’s suggestion) that perhaps it should be stated more explicitly. We now start our abstract with the sentence “Complex actions may be viewed as composites of behavioural modules, activated by specific modalities and operating in its characteristic timescales “. We hope that this sentence encapsulates the broad relevance of the manuscript, without overstating our case.

Reviewers' Comments:

Reviewer #2:

Remarks to the Author:

The authors have tried to address my major concerns with some significant changes to the text that now make their points come across more clearly (and with relevant literature cited to support their claims). My minor concerns are sufficiently addressed. Together, with the changes recommended by the other reviewers, the authors appear to have made their best effort to strengthen the paper for publication and sufficiently addressed my concerns.

Reviewer #3:

Remarks to the Author:

I feel that my comments/concerns have all been addressed. I had recommended an analysis that the seemed to me to be low-hanging fruit that could strengthen the narrative of the manuscript. The authors tried the analysis (and provided a reference of prior work that had tried to address the same question) with inconclusive results. But they've added text to the manuscript that makes it clear that they've explored that question and will allay readers like myself from thinking, "why didn't they just try this one thing?!"

Response to Reviewers' comments for NCOMMS-18-38698B

We are glad the reviewers find that our changes addressed their concerns and we deeply appreciate the thoughtful feedback provided by all three reviewers.

REVIEWERS' COMMENTS:

Reviewer #2 (Remarks to the Author):

The authors have tried to address my major concerns with some significant changes to the text that now make their points come across more clearly (and with relevant literature cited to support their claims). My minor concerns are sufficiently addressed. Together, with the changes recommended by the other reviewers, the authors appear to have made their best effort to strengthen the paper for publication and sufficiently addressed my concerns.

Reviewer #3 (Remarks to the Author):

I feel that my comments/concerns have all been addressed. I had recommended an analysis that seemed to me to be low-hanging fruit that could strengthen the narrative of the manuscript. The authors tried the analysis (and provided a reference of prior work that had tried to address the same question) with inconclusive results. But they've added text to the manuscript that makes it clear that they've explored that question and will allay readers like myself from thinking, "why didn't they just try this one thing?!"